# Adenine base editing efficiently restores the function of Fanconi anemia hematopoietic stem and progenitor cells

Sebastian M. Siegner[1,3], Laura Ugalde[2,3], Alexandra Clemens[1,3], Laura Garcia-Garcia[2], Juan A. Bueren [2], Paula Rio [2] ✉, Mehmet E. Karasu[1] ✉ & Jacob E. Corn [1] ✉

Fanconi Anemia (FA) is a debilitating genetic disorder with a wide range of severe symptoms including bone marrow failure and predisposition to cancer. CRISPR-Cas genome editing manipulates genotypes by harnessing DNA repair and has been proposed as a potential cure for FA. But FA is caused by deficiencies in DNA repair itself, preventing the use of editing strategies such as homology directed repair. Recently developed base editing (BE) systems do not rely on double stranded DNA breaks and might be used to target mutations in FA genes, but this remains to be tested. Here we develop a proof of concept therapeutic base editing strategy to address two of the most prevalent *FANCA* mutations in patient hematopoietic stem and progenitor cells. We find that optimizing adenine base editor construct, vector type, guide RNA format, and delivery conditions leads to very effective genetic modification in multiple FA patient backgrounds. Optimized base editing restored FANCA expression, molecular function of the FA pathway, and phenotypic resistance to cross-linking agents. ABE8e mediated editing in primary hematopoietic stem and progenitor cells from FA patients was both genotypically effective and restored FA pathway function, indicating the potential of base editing strategies for future clinical application in FA.

Fanconi Anemia (FA) is a serious genetic disorder mainly characterized by developmental abnormalities, cancer predisposition and bone marrow failure (BMF), a syndrome which becomes evident in most FA patients during the first decade of life[1–3]. Androgen therapy and regular blood transfusions can ameliorate the BMF in FA patients, but do not address the underlying cause of the disease[4]. Allogeneic hematopoietic stem cell transplantation (HSCT) from healthy donors is currently the only curative treatment of BMF in these patients, but carries serious risks such as graft-vs-host disease and increased incidence of squamous cell carcinoma in the long term[5,6]. Genetic treatments to complement or repair the mutations that cause FA during autologous HSCT would offer many benefits over traditional therapy, potentially providing a lasting cure without the side effects associated with allogeneic BMT[7].

FA is caused by mutations in any of the 22 genes that encode for proteins participating in the FA/BRCA DNA damage response pathway[8]. The FA gene products work together in physical complexes and connected pathways to repair interstrand cross links (ICLs) in DNA, which can be caused by DNA damaging agents such as chemotherapeutics (cisplatin, mitomycin C) or endogenous metabolic

[1]Department of Biology, ETH Zurich, Zurich, Switzerland. [2]Division of Hematopoietic Innovative Therapies, Centro de Investigaciones Energéticas Medioambientales y Tecnológicas and Centro de Investigación Biomédica en Red de Enfermedades Raras (CIEMAT/CIBERER) and Advanced Therapies Unit, Instituto de Investigación Sanitaria Fundación Jiménez Díaz (IIS-FJD, UAM), Madrid, Spain. [3]These authors contributed equally: Sebastian M. Siegner, Laura Ugalde, Alexandra Clemens. ✉e-mail: paula.rio@ciemat.ed; karasum@ethz.ch; jacob.corn@biol.ethz.ch

byproducts such as aldehydes[9,10]. In the absence of a functional FA pathway, these unresolved ICLs eventually lead to chromosomal breaks and genome instability. FA patient cells are also compromised by normal levels of pervasive stressors such as replication fork collapse[11], emphasizing the importance of the FA pathway to guardian genome integrity.

Lentiviral mediated gene therapy combined with optimized HSC mobilization and transduction protocols has been successfully used to treat HSCs from *FANCA*-deficient (FA-A) patients, the most prevalent FA complementation group[12,13]. Although no severe side effects have been observed in the FA gene therapy trial, the possibility of precise gene repair in mutated sequences offers an additional safeguard for FA HSCs in particular. Furthermore, gene editing would also maintain endogenous regulation of gene expression and could extend therapeutic application to other FA complementation groups.

"Classic" CRISPR-Cas genome editing relies on creating a targeted DNA double-strand break (DSB) that can be resolved by either error-prone pathways to create semi-random small insertions and deletions (indels), or by precise homology directed repair (HDR) from a template[14,15]. Although HDR may theoretically "surgically" replace almost any mutation to the wild type sequence, its efficiency is low in primitive HSCs and is particularly compromised in FA-HSCs due to their defects in HDR[16]. Indel-based genome editing has been demonstrated to be a good alternative to correct specific FA mutations, converting nonsense to in-frame mutations that restore the FA gene function[17]. This approach has, however, marked limitations in the spectrum of FA mutations that can be repaired.

Newer genome editing systems such as base editing (BE) that work without inducing double-strand DNA breaks theoretically offer great opportunities to precisely correct specific mutations in the FA genes[18,19]. Nevertheless, whether a path to a genetic cure for FA is feasible while using one of the many existing base editors is unclear. Here we report a BE approach to address two prevalent *FANCA* mutations in patient cells. We found that adenine base editing shows a remarkable efficacy to target FA alleles. Moreover, optimizations of the adenine base editor construct, vector type, guide RNA format, and delivery conditions restored FANCA expression, molecular function of the FA pathway, and phenotypic resistance to crosslinking agents. Importantly, ABE8e induced extremely high levels of gene conversion in hematopoietic stem and progenitor cells from healthy donors and FA patients, confirming the potential of this strategy for the future clinical application in FA.

## Results

### Optimization of adenine base editing conditions for editing FA deficient LCLs

To develop a proof-of-concept base editing therapy for FA, we first employed lymphoblastoid cell lines (LCLs) generated from either healthy donors (HD) or FA patients (Fig. 1a). These immortalized cells recapitulate the major phenotypic hallmarks of FA, including reduced proliferation and sensitivity to DNA crosslinking agents, and allowed us to test the efficacy and toxicity of different tools and protocols of BE. Mutations in *FANCA* account for approximately 60–65% of FA[20], and we focused on two prevalent alleles of *FANCA*[21,22]. FA-75 harbors an compound heterozygous mutation in *FANCA* gene (c.2639 G > A and c.3788_3790 del TCT)[17] while FA-55 carries a homozygous mutation in FANCA gene (c.295 C > T)[23].

Both the FA-55 and FA-75 mutations are not amenable to cytosine base editing (CBE) due to the identity and context of the mutations, but they could theoretically be addressed with adenine base editing (ABE) (Fig. 1a). The FA-75 G-to-A mutation could be reverted back to wildtype by targeting the adenine mutation in the coding strand. The FA-55 C-to-T mutation might also be reverted to wildtype by ABE targeting on the non-coding strand, although this mutation is very close to several other non-wobble coding strand thymidines that lie in the

base editing window (Fig. 1a). Modification of these positions would lead to coding changes expected to impair protein function. Therefore, we focused on editing the coding strand, in which fewer potentially deleterious bystander mutations could occur. Our strategy aimed to convert the FA-55 nonsense mutation to a tryptophan missense mutation. The targeted amino acid of FANCA is particularly non-conserved (Fig. 1a, Supplementary Fig. 2) and expendable for FANCA function[17]. Furthermore, the exact missense mutation is found in two otter species (Supplementary Fig. 2), suggesting that this change might be tolerated for FANCA function. We used PnB Designer to design several candidate gRNAs to base edit each FA genotype[24].

Base editors have rapidly diversified and multiple next-generation ABEs are available. ABEmax is a second-generation adenine base editor that has been used in several contexts and has been characterized extensively to establish its targeting window and off-target propensities[25–28]. We tested whether delivery of candidate gRNAs and ABEmax in plasmid format yielded intended base editing in FA-55 and FA-75 LCLs. Bulk Sanger sequencing and next generation Illumina sequencing of PCR amplicons (amplicon-seq) five days after electroporation of FA-55 and FA-75 LCLs indicated low levels of base editing (5.66 ± 0.59% A to G) for FA-55. In the case of FA-75, a 62.20 ± 2.12% of reads contained a G. However, FA-75 harbors a heterozygous mutation and so has a baseline wildtype level of 50% at the targeted sites (Fig. 1b, Sanger traces). These results suggested that these guide and base editor combinations were capable of genomic targeting to induce the desired sequence changes, though with modest efficiency in the current format.

To determine whether the respective edited alleles conferred proliferative advantage over cells harboring the mutant alleles, we monitored the growth of edited and unedited cells in bulk cultures using amplicon-seq that target each edited site. During 30 days of culture after editing, conversion of the FA-75 missense mutation to wildtype led to increased levels of the wild type base to 74.35 ± 6.35%. Significantly, conversion of the FA-55 nonsense mutation to missense also led to a proliferative advantage for edited cells, increasing to 29.67 ± 17.09% of the altered base (Fig. 1b, solid lines). Edited reads were not found in cells kept in culture for the same length of time but electroporated with only base editor and no gRNA, indicating that spontaneous reversion did not play a role in outgrowth of cells with the wildtype sequence (Fig. 1b, dashed lines).

Since double stranded DNA plasmid delivery is inefficient and highly toxic in HSCs[29], we next tested electroporation of an mRNA coding for ABEmax together with a chemically protected, synthetic gRNA. This combination has been effective in editing HSPCs from non-FA genotypes[30–32].

We found that ABEmax mRNA and synthetic gRNA dramatically increased bulk editing levels soon after electroporation for FA-55, with A to G substitution now contributing to the majority of the Sanger sequencing chromatogram (Fig. 1c) and no qualitative evidence of bystander mutations. mRNA based editing of the FA-75 allele was also improved relative to plasmid editing (Fig. 1c, right, bottom Sanger tracks), but was associated with a bystander mutation at the adjacent 3' adenine in the wobble position. We further quantified editing efficiency at each locus using amplicon-seq. mRNA delivery of ABEmax paired with synthetic guide RNAs yielded high levels of editing in both FA-55 (missense 53.14 ± 5.77% desired base and FA-75 correction, 74.75 ± 3.04% desired base) after 5 days in culture (Fig. 1c). In longer-term cultures, edited allele frequencies steadily increased for both *FANCA* genotypes, representing the great majority of reads after 30 days. We asked whether allele increase was due to functional correction or sustained mRNA expression by analyzing the frequency of the corrected SNP and a silent bystander mutation for FA-75. Alleles containing the correction and the silent bystander increased over time while reads containing only the bystander edit alone did not (Supplementary Fig. 3). This suggests

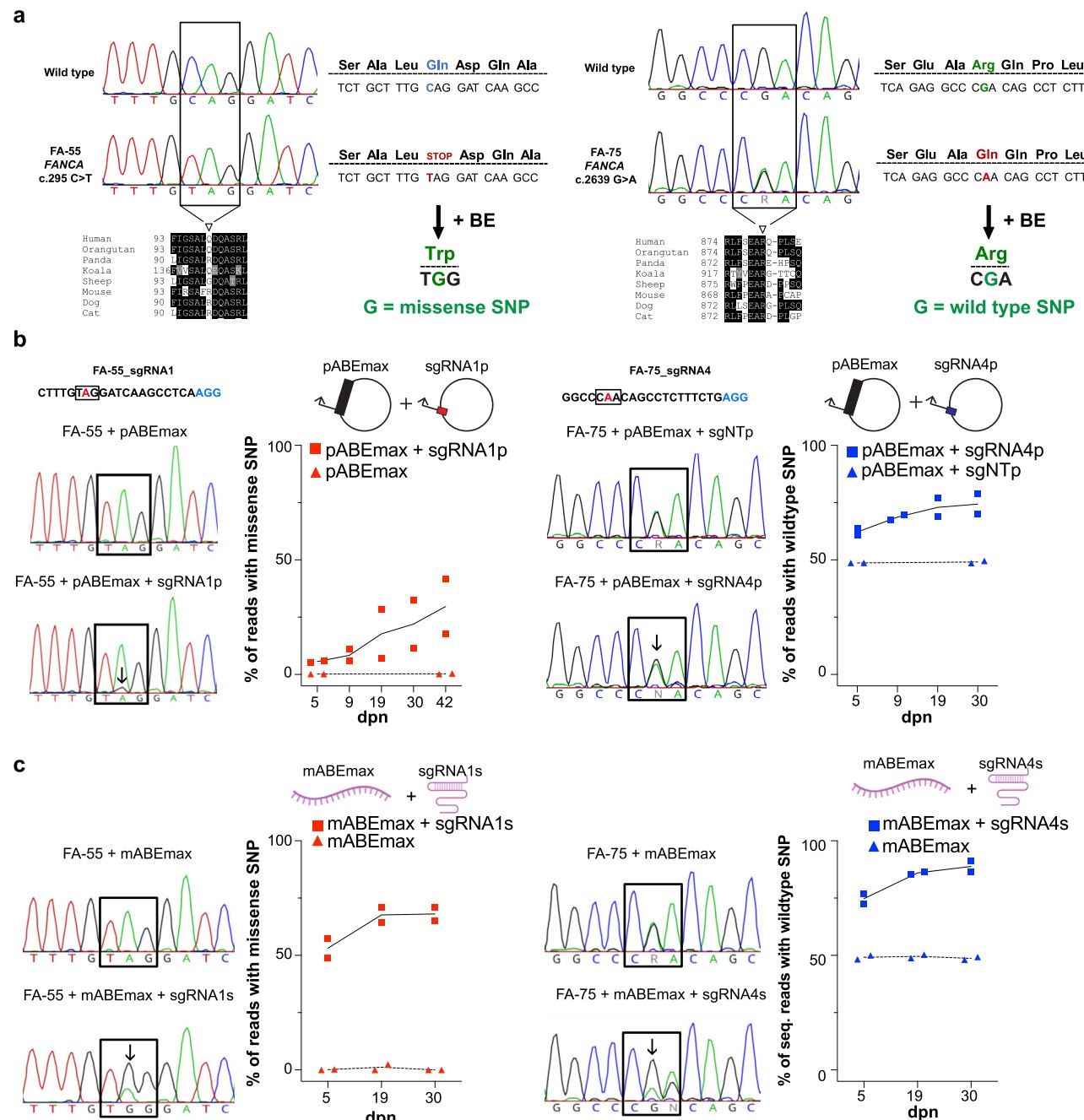

**Fig. 1 | Base editing is an efficient approach to modify Fanconi Anemia mutations. a** Details of FA-55 and FA-75 mutations. Sanger traces from wild type and mutant LCLs showed the indicated c.295 C > T and c.2639 G > A mutations. Next to Sanger traces, translation of codons is illustrated for each mutation. FA-55 c.295 C > T mutation leads to a stop codon, terminating translation of FANCA prematurely. Adenine base editing is designed to introduce a missense SNP that encodes a tryptophan instead of a glutamine. FA-75 c.2639 G > A leads to arginine to glutamine mutation. Adenine base editing reverts the missense SNP to wild type sequence. FA-75 is compound heterozygous and the wild type SNP is already present in unedited cells. Below Sanger traces, protein sequence alignments from multiple species are shown. Amino acids with dark background or grey background indicate identity or similarity among different species, respectively. **b** Base editing in FA-55 and FA-75 LCLs by delivering ABEmax and sgRNA in plasmid format. On the left top side, the FA-55 sgRNA1 target site is shown. PAM sequence and edited base are highlighted by blue and red fonts, respectively. The FA-75 sgRNA4 site is shown. Representative Sanger traces show initial editing 5 days after electroporation of FA-55 and FA-75 LCLs, all with ABEmax and with or without the indicated sgRNAs. In the case of FA-75 targeting, plasmid carrying non targeting sgRNA (sgNTp) was included. Arrows indicate the presence of edited alleles. Graphs show edited allele frequency measured by amplicon NGS in a time course after editing. Solid shapes represent the pool of cells electroporated with base editor and sgRNA, while triangle shapes represent the pool of cells electroporated with base editor alone. The continuous lines and dashed lines summarize 2 biological replicates. **c** Base editing of FA-55 and FA-75 LCLs by delivering ABEmax and sgRNA in mRNA format. Representative Sanger traces show initial editing 5 days after electroporation with ABEmax with or without sgRNAs. Arrows indicate presence of edited alleles. Graphs show edited allele frequency measured by amplicon NGS in a time course after editing. Solid shapes represent the pool of cells electroporated with base editor and sgRNA, while triangles shapes represent the pool of cells electroporated with base editor alone. The continuous lines and dashed lines summarize 2 biological replicates. Source data are provided as a Source Data file.

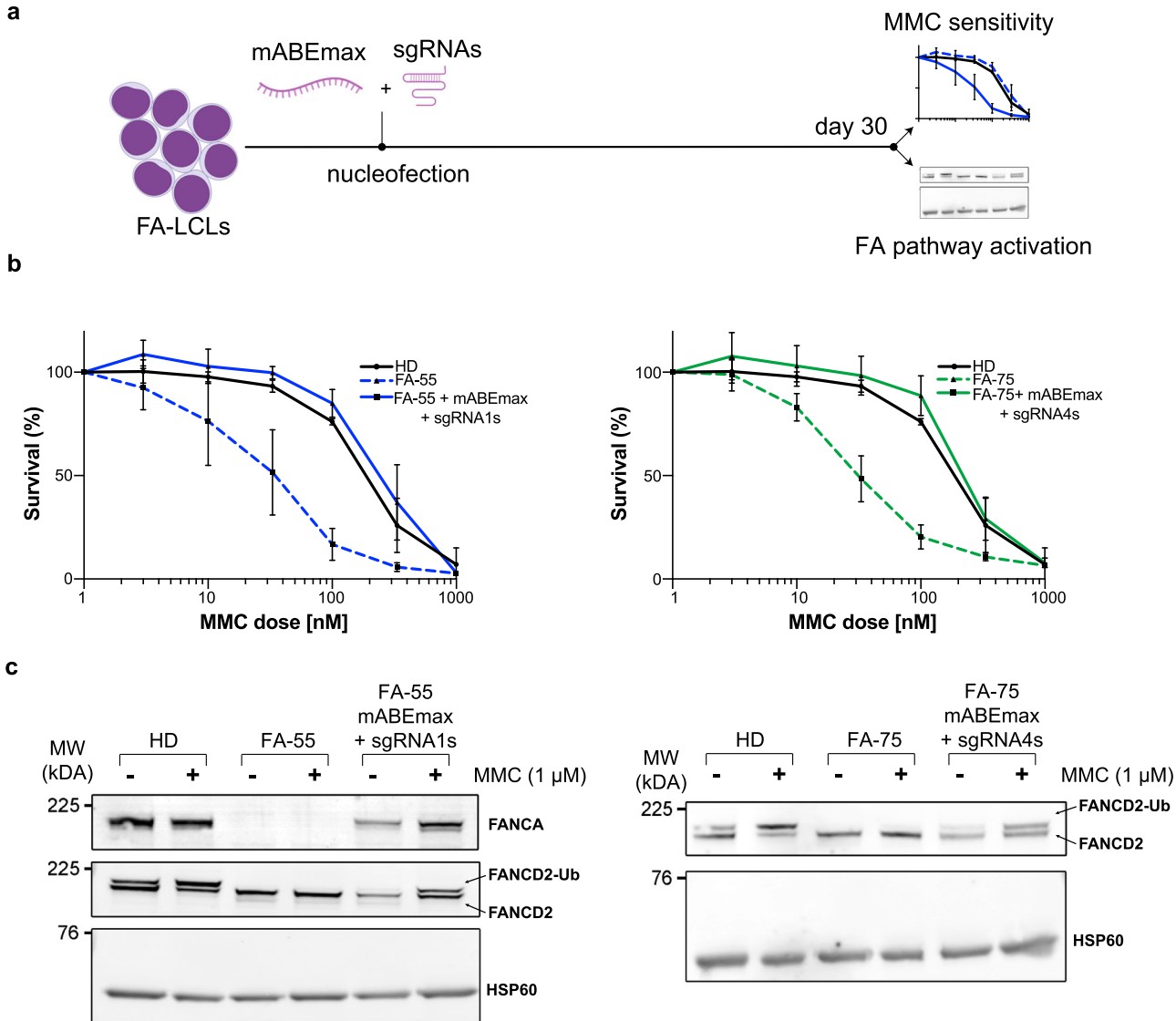

**Fig. 2 | Base editing successfully reverts classical FA phenotypes. a** Schematics of experimental design to edit FA-55 or FA-75 LCLs. Cells were edited with base editor mRNA and synthetic gRNA and grown for 30 days in culture. Editing efficiencies were assessed by amplicon NGS, as shown in Fig. 1c. (Created with BioRender.com). **b** MMC resistance of edited FA-A LCLs. Black line indicates healthy donor (HD) LCL response to increasing doses of MMC. Dashed blue or green lines represent FA-55 or FA-75 LCLs, respectively. Solid blue or green lines represent FA-55 or FA-75 edited pools, respectively. The graphs summarize the mean of 3 biological replicates, error bars indicate SD. **c** Representative Western blots. Indicated cell populations were challenged with 1 µM MMC for 24 h. Protein extracts were analyzed by Western blot with indicated antibodies (anti-FANCA, anti-FANCD2, anti-HSP60). FANCD2 or FANCD2-Ub bands are indicated by arrows. ($n$ = 2, biologically independent experiments). Source data are provided as a Source Data file.

that the increase of corrected alleles is due to the proliferative advantage of corrected cells. Taken together, these results indicate that using mRNA delivery of ABEmax paired with a synthetic guide RNA can be very effective at genetic modification in two different *FANCA* genotypes. Our results also suggest that the *FANCA* missense edit we tested here is capable of providing a fitness benefit relative to cells with FA-55 c.295 C > T mutation.

## ABE edited FA LCLs have restored functional FA pathway
We next asked if the base edited LCLs have restored FA pathway function (Fig. 2a). Cells derived from FA patients exhibit hypersensitivity to DNA interstrand crosslinking reagents such as mitomycin C (MMC) and cisplatin[33]. Unedited FA-55 and FA-75 LCLs were both hypersensitive to MMC compared to HD LCLs (Fig. 2b). To test if gene editing modified the MMC-hypersensitivity of these cells, samples were electroporated with ABEmax mRNA and synthetic guide RNAs,

passaged for thirty days in culture to expand corrected cells, and then assayed for MMC sensitivity. At this extended time point, both FA-55 and FA-75 LCLs exhibited complete phenotypic restoration (Fig. 2b).

FANCD2 monoubiquitination is a molecular hallmark of FA pathway activation in response to MMC exposure[34]. In the absence of functional FANCA protein and FA core complex assembly, the FANCD2-FANCI heterodimer cannot be monoubiquitinated. In HD LCLs we verified robust basal expression of FANCA and MMC-induced monoubiquitination of FANCD2 (Fig. 2c). Neither the FA-55 nor FA-75 LCLs were capable of monoubiquitinating FANCD2 in response to MMC treatment. However, bulk ABEmax edited pools expanded for 30 days and robustly ubiquitinated FANCD2 after MMC exposure (Fig. 2c). Notably, the missense edit conferred to the FA-55 LCL restored FANCA protein expression and FANCD2 monoubiquitination, further highlighting that the nonsense-to-tryptophan base edit was sufficient to rescue the FA pathway.

## ABE8e increases editing efficiency but not phenotypic correction of FA LCLs

While we were in the process of characterizing ABEmax-edited FA LCLs, a hyperactive adenine base editor variant was developed by the labs of Jennifer Doudna and David Liu[35]. ABE8e was reported to outperform ABEmax in terms of editing efficiency in some cell lines, but with a slightly increased propensity for bystander and off-target effects. We wondered whether ABE8e could further increase base editing levels at early timepoints, especially in FA patient backgrounds, since achieving a very high level of initial editing could be critical when attempting to edit the especially rare HSPCs that can be isolated from FA patients.

To compare the efficiencies of ABE8e and ABEmax, we followed a similar experimental design as described in Fig. 2. To determine whether the hyperactive ABE8e was more efficient to generate point conversions without relying on a survival advantage of edited cells, we amplicon-sequenced cell pools just five days after electroporation (Fig. 3a). ABEmax showed good editing efficiency (36.43 ± 16.21% and 73.70 ± 8.74% in FA-55 and FA-75, respectively), but ABE8e even further increased this efficiency to 72.31 ± 10.82% and 89.68 ± 7.07%, respectively (Fig. 3b, c). Concomitant to this effect, we detected higher bystander editing in FA-75 cells edited when ABE8e was used, compared to ABEmax edited (Supplementary Fig. 4). In the case of FA-55, only a bystander edit was observed when ABE8e used for the editing, albeit less than <5% of total reads (Supplementary Fig. 4).

Since the correction of a single allele is sufficient to correct the disease phenotype in FA[36,37], we asked whether ABE8e edited pools exhibited greater phenotypic correction at short time points than ABEmax edited pools. We thus tested MMC sensitivity and FANCD2 monoubiquitination only nine days after editing (Fig. 3a). Despite higher editing by ABE8e, we found that both ABE8e and ABEmax yielded similar levels of MMC resistance, which was equivalent to those observed in HD LCLs (Fig. 3d). The phenotypic correction of FA-55 and FA-75 LCLs was also supported by the restoration of FANCD2 ubiquitination in both the ABEmax and ABE8e edited pools (Fig. 3e).

## Assessment of potential off-target editing for FA-55 sgRNA1 or FA-75 sgRNA4

Cas-based genome editing tools can affect off-target genomic loci that have sequences similar to the on-target guide RNA[38]. To further characterize ABE8e and ABEmax editing in FA LCLs, we computationally predicted potential off-target sites for both the FA-55 and FA-75 guide RNAs using Cas-OFFinder[39] and CRISTA[40]. For the FA-55 targeting sgRNA1 we analyzed 38 potential sites (Supplementary Table 1). For the FA-75 targeting sgRNA4 we tested 29 potential sites (Supplementary Table 2). The FA-55 targeting sgRNA1 exhibited no detectable editing in any of the tested candidate off-target sites, irrespective of base editor (Fig. 4a). However, the FA-75 targeting sgRNA4 proved to be more susceptible to off-targeting editing, especially when combined with ABE8e. One prominent off-target site (OT3) was located on chr2 in the intron 13 of an uncharacterized gene named *KIAA2012*, with 21.14 ± 4.16% base editing with ABE8e and 4.93 ± 3.23% with ABEmax (Fig. 4b). With ABE8e we also found low levels of off-target editing at OT11 (~3% editing), OT16 (~2% editing) and OT27 (~1.5% editing) (Fig. 4b). OT11 is located in an intergenic region, OT16 is in intron 1 of *TRIO*, and OT27 is in intron 1 of *ZNF267*. The potential effects of low-level editing at these non-coding off-targets remain to be determined. Overall, our results indicate that the FA-55 targeting gRNA has no OTs at the tested sites with either base editor. The FA-75 targeting gRNA has some OTs in intergenic and intronic sites that are increased by the use of ABE8e.

## Long-term HSPCs can be edited using ABE8e

Given the promising results obtained in immortalized patient cells, we asked whether base editing approaches would potentially be suitable in preclinical models and primary cells. Before moving to precious HSPCs from FA patients, we optimized electroporation conditions in HD CD34+ cells by targeting the *AAVS1* safe harbor locus with both ABEmax and ABE8e. We electroporated varying amounts of CD34+ cells from cord blood (CB) and mobilized peripheral blood (mPB) with mRNA forms of each base editor and a synthetic guide RNA targeting *AAVS1* (Fig. 5a and Supplementary Fig. 6a)[41]. As in LCLs, ABE8e was much more efficient than ABEmax in both CB CD34+ cells (85.6 ± 1.7% ABE8e vs 30.7 ± 5.9% ABEmax) and the more clinically relevant mPB CD34+ cells (71.2 ± 13.30% ABE8e vs 28.0 ± 6.2% ABEmax) (Fig. 5b, c).

Analysis of individual hematopoietic colonies showed that ABE8e generated point conversions in homozygosis in all cases, confirming its efficiency in HSPCs from CB and mPB (Supplementary Fig. 6d, e). However, NGS performed in cells maintained in liquid cultures also revealed 4.2 ± 1.3% bystander mutations in the targeted locus when ABE8e was used. Regardless of the base editor, electroporation of base editor with synthetic guides into purified CD34+ cells did not cause gross defects in the cell viability and clonogenic potential of the HSPCs, suggesting that base editing was well tolerated in these cells[42](Supplementary Fig. 6b–e).

To confirm that base editing can efficiently target long-term repopulating HSCs and does not affect the engraftment capacity of these precursors, unedited and edited CD34 + cells from CB and mPB sources from HDs were serially transplanted into NOD.Cg-*Prkdc*scid *Il2rg*tm1Wjl/SzJ (NSG) immunodeficient mice (Fig. 5a). Monthly post-infusion, BM cells were collected from transplanted recipients by femoral BM aspiration. Three months post infusion mice were sacrificed and bone marrow cells were transplanted in secondary recipients. Human engraftment was analyzed by flow cytometry using anti-hCD45. Multilineage reconstitution was assessed using anti-hCD34-APC, anti-hCD33-PE, anti-hCD19-Pe-Cy5, and anti-hCD3-Pe-Cy7 and to discard pre/pro B cells from CD34+ cells, hCD34+ CD19- were also analyzed.

We found that CB HSPCs edited with either ABEmax or ABE8e engrafted with similar efficiencies as compared to mock-edited counterparts both in primary (median levels of engraftment 68.5 ± 14.6% mock; 69.6 ± 11.9% ABEmax and 72.0 ± 9.7% ABE8e) and secondary recipients 18.8 ± 13.8% mock; 18.4 ± 15.3% ABEmax; 20.4 ± 10.6% ABE8e (Fig. 5d, e), confirming that base editing by ABE8 did not affect the engraftment of LT-HSCs. As expected, engraftment of mPB CD34+ cells was lower than for CB, but also comparable between mock and base edited cells (31.4 ± 25.4% mock; 40.5 ± 23.3% ABEmax; 42.9 ± 17.0% ABE8e) (Fig. 5g). No overt toxicity associated to the treatment was observed in these mice (Fig. 5d, e). Also the proportion of donor myeloid and lymphoid lineages of hCD34+ cells present in BM from transplanted recipients were similarly represented in groups transplanted with unedited and the two type of edited cells (Supplementary Fig. 6f–h).

Amplicon sequencing analysis in edited hCD34+ cells that engrafted recipient mice showed the higher efficacy of ABE8e compared to ABEmax, reaching median values of editing higher than 50% at 3-4 months post-transplant, regardless of the HSC source (Fig. 5f, h). High editing efficiencies were also observed at three months post-transplantation in secondary recipients that had been infused with CB ABE8e hCD34+ cells, highlighting the potential of ABE8e to target long-term repopulating HSCs.

## Efficient editing of FA patient HSPCs with ABE8e

Finally, we investigated whether base editing approaches could correct mutations in HSPCs from FA patients. Because of the extreme scarcity of HSPCs in FA patients, we tested the efficiency of gene editing in Lineage depleted (Lin-) cells from one patient and CD34+ enriched cells from three different FA patients, all carrying the *FANCA* c.295 C > T mutation as in FA-55 LCLs (Fig. 6). Since ABE8e editing of FA-55 LCLs exhibited the highest on-target activity with low level of bystander

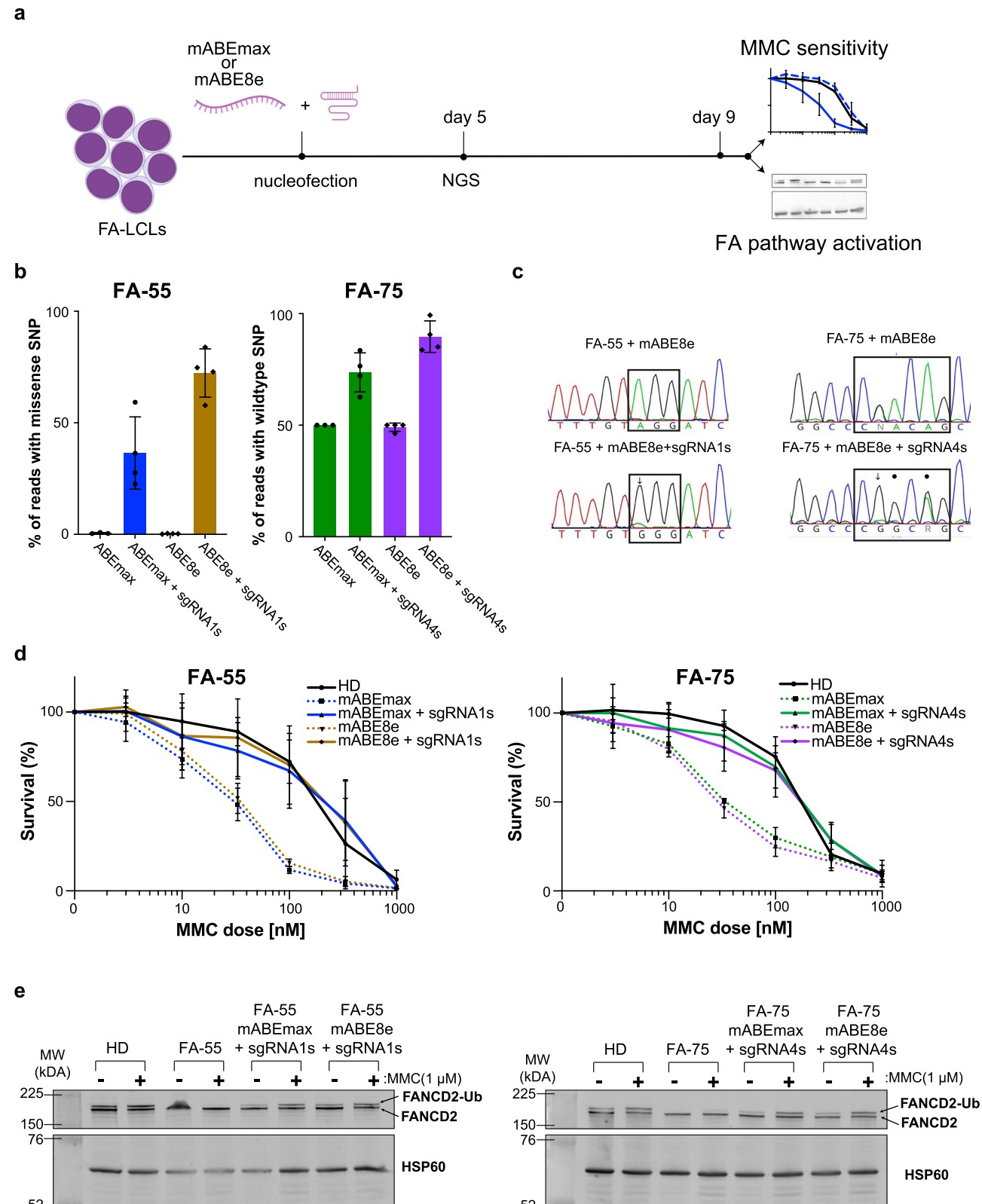

modifications and no off-target activity in LCLs (Fig. 3b), this base editor was selected to target FA patient HSPCs.

Lin⁻ and CD34⁺ enriched cells obtained from thawed mPB samples were pre-stimulated for 24 h and electroporated with ABE8e mRNA together with the sgRNA1, previously used in FA-55 LCLs. 24 h later cell survival was analyzed by flow cytometry in mock and ABE8e edited cells (Fig. 6a). This analysis confirmed the toxicity associated with the

use of electroporation of thawed FA-A HSPCs with variable survival rates depending on the patient sample which varied from 22.5% up to 59.9% when CD34⁺ enriched cells were used. Due to the number of cells available, electroporation without base editor was used as a control. Clonogenic assays conducted in the absence and the presence of MMC performed in patient 1 Lin⁻ cells and patient 1 and 3 CD34⁺ enriched cells showed high levels of MMC resistance (137.5%, 80.8% and 60.0%,

**Fig. 3 | Comparison of ABEmax and ABE8e for editing and phenotypic correction in FA LCLs. a** Schematics of experimental design to edit FA-55 or FA-75 LCLs by ABEmax or ABE8e using mRNA base editor and synthetic gRNAs. Electroporated cells were collected at day 5 to measure editing efficiency and at day 9 to measure MMC resistance and activation of FANCD2 monoubiquitination. (Created with BioRender.com). **b** Quantification of editing levels by amplicon NGS on day 5 in edited LCL populations. Dot or rhombus indicate individual experiments, bars represent the mean of 4 independent experiments and error bars indicate SD. **c** Representative Sanger traces show initial editing 5 days after electroporation of FA-55 and FA-75 with ABE8e mRNA and with or without indicated synthetic sgRNAs. Arrows indicate the presence of edited alleles. Dots indicate the presence of bystander edits. **d** MMC survival of edited FA-A LCLs. Black lines indicate healthy donor (HD) LCL response to increasing doses of MMC. Dashed lines represent FA-55 or FA-75 LCLs electroporated only with ABEmax or ABE8e. Solid colored lines represent FA-55 or FA-75 electroporated with sgRNAs and ABEmax or ABE8e. The graphs summarize the mean of 4 (FA-55) and 3 (FA-75) biological replicates, error bars indicate SD. **e** Representative Western blots. Indicated cell populations were challenged with 1 μM MMC for 24 h. Protein extracts were analyzed by Western blot with indicated antibodies (anti-FANCD2, anti-HSP60). FANCD2 or FANCD2-Ub bands are indicated by the arrows. ($n = 2$, biologically independent experiments). Source data are provided as a Source Data file.

respectively) (Fig. 6b), confirming the restoration of the FA pathway after ABE8e editing. Notably, the number of CFCs/$1 \times 10^5$ cells in CD34$^+$ enriched cells was very similar in mock and ABE8e edited cells, confirming that BE per se is not affecting the clonogenic capacity of FA HSPCs (Supplementary Fig. 7a).

The analysis of the gene editing efficiency at 5 days post electroporation confirmed high editing levels both in Lin$^-$ cells from patient 1 (43.06%) and also in CD34$^+$ enriched cells from patient 1 (57.51 ± 21.00%), patient 2 (64.37%) and patient 3 (42.22%) (Fig. 6c). When individual colonies were analyzed 14 days after plating, we observed higher editing efficiency of edited allele indicating a proliferative advantage of edited cells over non edited cells (Fig. 6d). Sanger sequencing of individual colonies showed that gene editing occurred at both alleles in most of the colonies analyzed from the different FA HSPCs (Supplementary Fig. 7b). To further demonstrate the safety of editing FA HSPCs, we analyzed off target editing at OT1-8 and OT36 in patient 2 and 3. Nevertheless, we did not detect any elevated A to G conversion in the ABE8e edited samples at these off-target sites (Supplementary Fig. 8).

Finally, to analyze the self-renewal and differentiation capacities of HSPCs from mock and ABE8e edited CD34$^+$ enriched cells (from patient 2), primary colonies grown in semisolid cultures were collected and replated in additional methylcellulose cultures. Only ABE8e edited colonies could generate secondary colonies, confirming that correction by ABE8e also improved the self-renewal and differentiation capacity in the FA hematopoietic progenitor cells (Fig. 6e, f).

Collectively our data demonstrates the high efficiency of ABE8e to target HSPCs from HDs and FA patients and highlights the potential of base editors to correct a prevalent mutation observed in FA.

## Discussion

Here we explored the possibility of using CRISPR-Cas base editors to reverse the effects of FA mutations in patient-derived LCLs and HSPCs. While NHEJ- and HDR-based strategies have been explored to genetically treat FA HSPCs[17,43–45], this is the first study to our knowledge that demonstrates proof-of-concept that base editors are tolerated and highly efficient in FA HSPCs.

The FA pathway is multi-functional, with roles in DNA crosslink repair, DSB repair and replication fork restart, among other relevant functions[46–48]. Lack of these activities has compromised HDR-based approaches for allele correction in FA patient cells. However, we found that absence of functional FA pathway did not interfere with adenine base editor activity in LCLs and HSPCs. This suggests that the FA pathway is dispensable for base editor activity and could be exploited as a novel therapeutic strategy in FA. Both *FANCA* nonsense-to-missense and missense-to-wildtype editing resulted in phenotypic rescue on both the molecular and phenotypic level. We are optimistic that the approach outlined here could be extended to additional FA alleles to form the foundation of future gene editing therapies for FA.

We found that ABE editing can yield phenotypic correction in FA LCLs and FA HSPCs. In edited FA-55 LCLs, we observed re-expression of FANCA protein, molecular evidence of FA pathway re-activation, and a significant proliferative advantage over unedited cells. Similarly, edited FA-75 LCLs showed phenotypic correction on multiple levels. Importantly these data were confirmed in FA HSPCs, where high level of base editing was observed when ABE8e was used. Furthermore, edited cells showed proliferative advantage, agreeing the previous findings[12,17]. Correction of the FA pathway in HSPCs was also confirmed by the MMC resistance observed in ABE8e edited HSPCs. A replating CFU assay in patient 2 CD34$^+$ enriched cells also highlighted the self-renewal and proliferation capacity of ABE8e edited cells versus mock unedited ones. However, the limited access to sufficient numbers of FA HSPCs prevented us from currently determining whether these phenotypically corrected FA HSPCs efficiently engraft in an immunodeficient mouse model. Such a test will be an important step in further preclinical studies. Importantly, in vivo experiments in immunodeficient mice confirmed that the engraftment capacity of base edited HSPCs was not altered in comparison with unedited cells, either from HD CB or mPB CD34 + cells. Amplicon-sequencing analysis in BM obtained from transplanted animals also confirmed that ABE8e can efficiently target LT-HSCs, a key requirement for the future application of this strategy in FA patients. These results are consistent with recent studies performed in healthy and sickle cell HSPCs[42,49,50]. Further studies will directly address the important question of engraftment potential for base edited FA cells and the best BE delivery system to minimize toxicity in FA HSPCs for the future application in FA patients.

The extremely high activity of ABE8e might result in higher levels of unintended mutations in the editing window and at off-target loci[35]. We detected high levels of unintended bystander mutations in the FA-75 editing window, but not at the FA-55 site. In FA LCLs we also found one prominent and three low frequency ABE8e off-targets for the FA-75 sgRNA, but no off-targets for the FA-55 sgRNA. Despite unintended modifications in ABE8e edited cells, it was capable of much higher editing efficiency in CD34$^+$ HSPCs (Fig. 5), thus enabling editing of clinically-relevant long term HSPCs. ABE8e could still be valuable in FA-75, since the bystander edits occur at wobble bases and are predicted to be neutral once the corrected allele expands. Our phenotypic analysis in FA-75 LCLs (Fig. 3) indicated that these bystander mutations do not affect FANCA function and FA pathway activation. If needed, one could reduce the bystander and off-target activities of ABE8e by using an ABE8e RNP or ABE8e virus-like particles, which have been reported to reduce off-target DNA effects[35,51]. The ABE8e (TadA-8e V106W) variant could also be a useful tool to reduce unintended base modification. Unbiased off-target identification to identify potential stereotyped edits for example using whole genome sequencing or RNA-seq before and after base editing will be an important next preclinical step in validating any base editing approach to cure FA.

Overall, our study indicates that adenine base editing is a feasible approach for the efficient restoration of function in FA patients' HSPCs. These results provide the basis for the use of base editors in FA and other DNA repair disorders, where these targeted tools may be both more efficacious and even safer, as compared to current untargeted gene addition therapies.

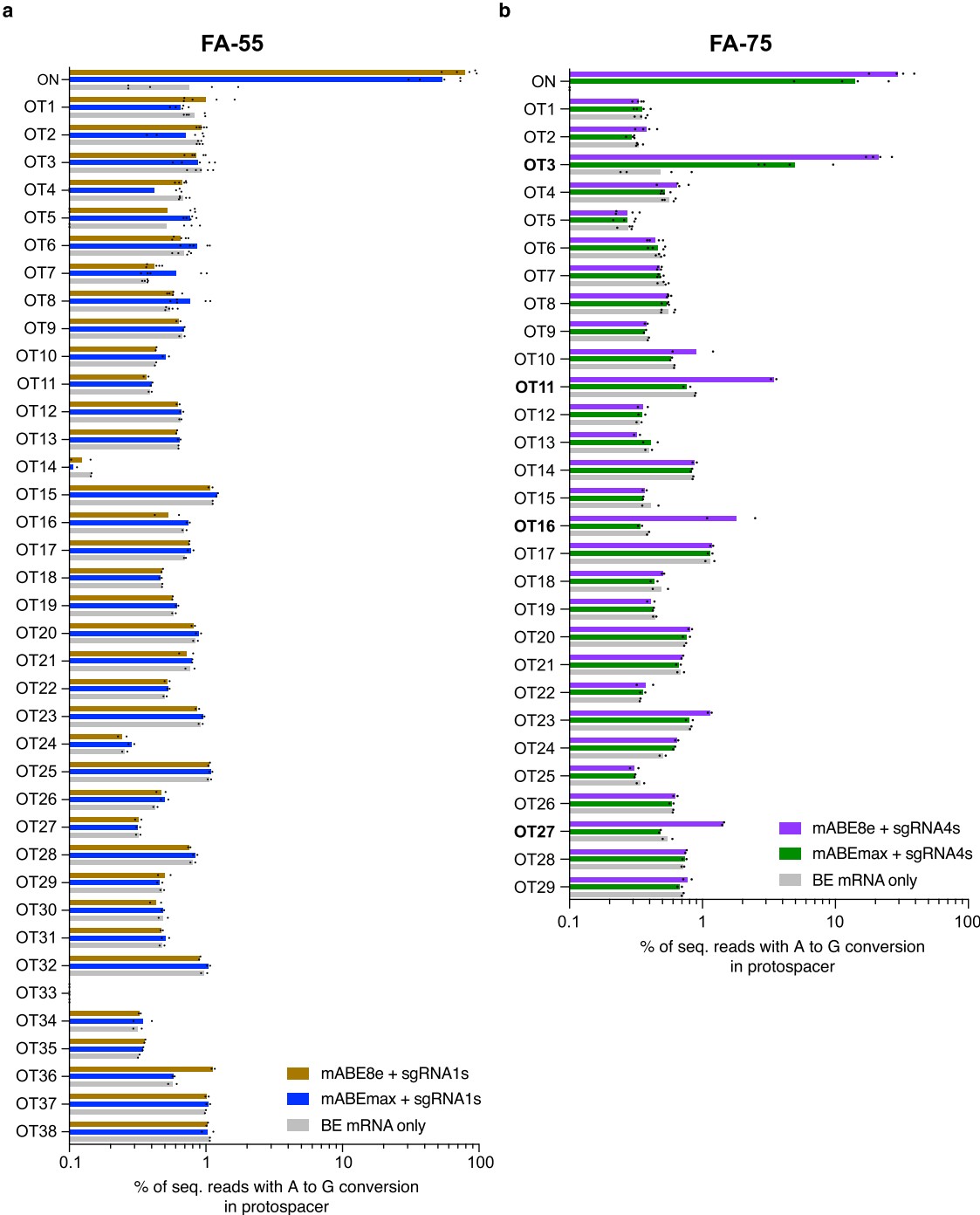

**Fig. 4 | Off-target analysis at predicted loci in FA-55 and FA-75 LCLs.** Computationally predicted off-target sites are shown for FA-55 sgRNA1 **a** and FA-75 sgRNA4 **b**. The Sum of all NGS reads containing one or more A to G conversion in the protospacer are plotted. Each biological replicate is shown by a dot and bars represent mean of amplicon NGS editing levels from all replicates. OT3, OT11, OT16, and OT27 are marked in bold, indicating potential off-targets for sgRNA4. For FA-55, 5 biological replicates are shown for the on-target site and OT1-8 while 2 biological replicates are shown for OT9-38. For FA-75, 4 biological replicates are shown for the on-target site and OT1-8 while 2 biological replicates are shown for OT9-29. Source data are provided as a Source Data file.

## Methods

### Ethical statement

All experimental procedures with mice were conducted according to European and Spanish regulations (European convention ETS 123, regarding the use and protection of vertebrate mammals used in experimentation and other scientific purposes, Directive 2010/63/UE, Spanish Law 6/2013 and *Real Decreto* (R.D.) 53/2013 regarding the protection and use of animals in scientific research). Procedures involving Genetically Modified Organisms were conducted according to the proper European and Spanish regulations (Directive 2009/41/CE, Spanish Law 9/2003, and R.D. 178/2004). Procedures were approved by the CIEMAT Animal Experimentation Ethical Committee according to all external and internal biosafety and bioethics guidelines and authorized by the *Comunidad de Madrid* Government (Codes: PROEX #070-15 and PROEX #156.5/21 Cell and Gene Therapy in rare diseases with chromosomal instability).

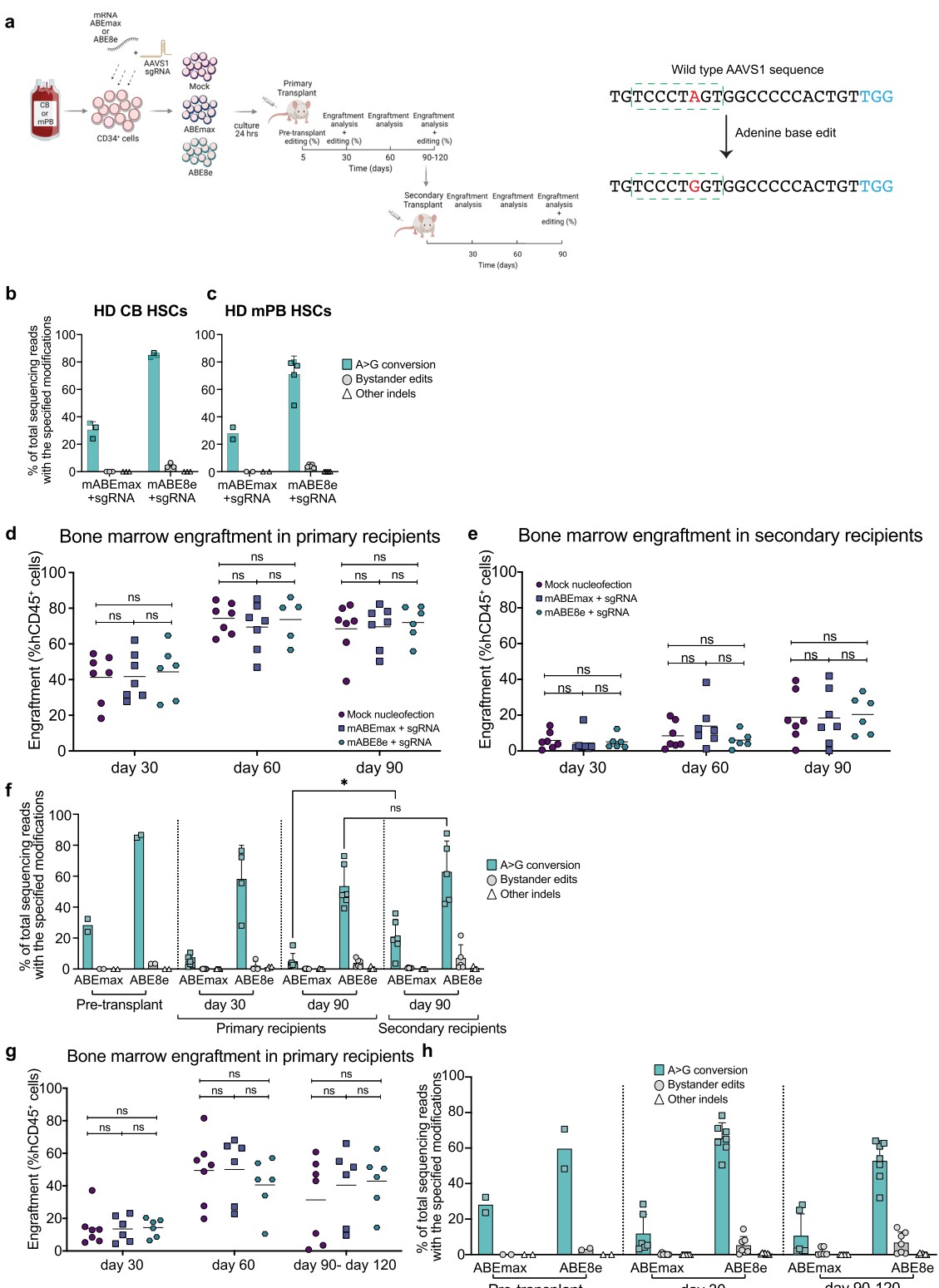

HD CB CD34+ cells were obtained upon approval by the *Centro de Transfusiones* de la *Comunidad de Madrid* (Madrid) and after informed consent was signed. The use of human mPB samples has been approved by the Ethics Committee at *Hospital Infantil Universitario Niño Jesús* (Madrid) (Ref: 07/029193.9/21). In this respect, HD CB derived cells from male and female were used (Fig. 5b, d, e, f and Supplementary Fig. 6a, b, d, f, g). HD mPB were obtained from n = 4 female donors and n = 1 male donor aged 14–39 (Fig. 5c, g, h and Supplementary Fig. 6c, e, h). mPB samples from FA-A patients under 5

years were obtained from the discarded negative fraction after CD34+ selection in the FANCOSTEM trial (Eudra number CT 2011-006197-88) (Fig. 6 and Supplementary Fig. 7). In all cases participants or legal representatives were informed in detail about the research purpose and informed consent was signed.

**Plasmid generation**

sgRNAs were designed to contain a variable 20 nucleotide sequence, corresponding to the target gene. Oligos for sgRNAs and nicking

**Fig. 5 | Adenine base edited human CD34⁺ cells successfully engraft into NSG mice. a** Experimental design followed to target CD34⁺ primary cells by base editing. Healthy donor CD34⁺ cells from cord blood (CB) or mobilized peripheral blood (mPB) were purified by immunoselection and pre-stimulated 24 hours prior to base editing with mABEmax or mABE8e in combination with an *AAVS1* targeting synthetic gRNA. Edited CD34⁺ were maintained in culture for 24 h and then transplanted into immunodeficient NSG mice. Secondary transplants were conducted in mice transplanted with CB HD CD34⁺ cells. Engraftment analyses were conducted at day 30, 60 and 90–120 post-transplantation in primary and secondary recipients. Amplicon NGS analysis was conducted 5 days after electroporation (pre-transplant) and 30 and 90–120 days after transplantation in primary and 90 days after transplantation in secondary recipients. AAVS1 PAM site is highlighted with light blue and green dashed rectangle indicates the potential editing window. Detailed bystander information can be found in Supplementary Fig. 9, even though bystander edit was outside of the canonical base editing window, out of caution, we noted as bystander edit. (Created with BioRender.com). **b, c** Base editing frequencies at *AAVS1* in CB **b** and mPB **c** CD34⁺ cells edited with mABEmax or mABE8e measured by amplicon sequencing. Bars represent mean ± SD values from 3 independent experiments with ABEmax and ABE8e in HD CB HSCs **b** and two and five independent experiments for ABEmax and ABE8e in HD mPB HSCs respectively **c**. **d, e** Human bone marrow engraftment of edited CD34⁺ cells from CB at 30, 60 and 90 days after transplant in primary **d** and secondary **e** recipients. Mean values are represented with a horizontal bar (number of mice analyzed: n = 7, n = 7 and n = 6 in primary and secondary recipients at 90 days for mock, ABEmax and ABE8e respectively). In all cases, a two-way ANOVA was performed followed by a Tukey's multiple comparison test: ns = not significant. **f** Amplicon sequencing analysis of editing levels in CD34⁺ CB cells pre-transplant at 5 days post electroporation (n = 2, data correspond to cells shown in Fig. 5b) and 30 and 90 days post transplantation from primary recipients (number of mice analyzed: n = 7 and n = 4 at 30 days; n = 6 and n = 6 at 90 days for ABEmax and ABE8e respectively) and 90 days post transplantation from secondary recipients (number of mice analyzed: n = 6 and n = 5 for secondary recipients for ABEmax and ABE8e respectively). Bars represent mean value and error bars represent SD. **g** Human bone marrow engraftment of edited CD34⁺ cells from mPB 30, 60 and 90 or 120 days after transplant. Mean values are represented with a horizontal bar (number of mice analyzed: n = 7, n = 6 and n = 6 for mock, ABEmax and ABE8e respectively). In all cases, a two-way ANOVA was performed followed by a Tukey's multiple comparison test: ns = not significant. **h** Amplicon NGS analysis of editing levels in CD34⁺ mPB cells pre-transplant at 5 days post electroporation (n = 2, data correspond to cells shown in Fig. 5c) and 30 and 90–120 days post transplantation from primary recipients (number of mice analyzed: n = 7 and n = 4 at 30 days; n = 6 and n = 7 at 90–120 days, for ABEmax and ABE8e respectively). Bars represent mean values and error bars represent SD. Source data are provided as a Source Data file.

guides were ordered from IDT and cloned into the pLG1-puro-BFP vector after digestion with BstXI and BlpI. Base editor plasmid ABEmax_P2A_GFP (Addgene plasmid # 112101) was a gift from David Liu and Lukas Villiger. The coding sequences of ABEmax or ABE8e were cloned into a T3 promotor containing pRN3 plasmid using the NEBuilder® HiFi DNA Assembly Master Mix (New England Biolabs). All vectors were purified using Qiagen Spin Mini- or Midiprep (Qiagen) with endotoxin removal step. Primers used in this study can be found in Supplemental Table 3.

## mRNA production for ABE base editors
All mRNA used in this study was generated by the following synthesis protocol. The mRNA template plasmid was linearized by digestion with SfiI (50 °C, overnight) and 200 μl of the digestion reaction was combined and mixed 1:1 with phenol chloroform for extraction. Samples were vortexed for 15 s. at high speed and then centrifuged at $13,000 \times g$ for 5 min. A total of 150 μl of the aqueous phase were transferred into a new tube and 1:10 volume of 3 M NaOAc and 165 μl of isopropanol were added. After 30 min incubation at −80 °C the samples were centrifuged at 4 °C, top speed for 30 min. The supernatant was carefully removed while not disturbing the pellet and 400 μl of 80% EtOH were added for another spin of 5 min. The EtOH was removed without leaving residuals and the pellet was dissolved in 10 μl of RNAse-free water. For in vitro transcription, the mMESSAGE mMACHINE® T3 Kit (Life technologies) was used as described in the manual. A total of 1 μg of linear plasmid was used as a template and transcription reaction was carried out for 2 h at 37 °C. For removal of the residual DNA template, 1 μl of TURBO DNAse was added to the transcription reaction for 15 min. RNeasy Mini Kit (Qiagen) was used for cleanup of the transcription reaction. In vitro transcribed mRNAs were kept at −80 °C until further use.

## Cell lines
Patient-derived LCLs (FA-55 and FA-75) and HD-LCLs were a gift from Dr. Paula Rio, (CIEMAT, Spain). LCLs were cultured in Roswell Park Memorial Institute medium (RPMI from Thermo Fisher Scientific) supplemented with 20% Hyclone fetal bovine serum (FBS), 1% penicillin/streptomycin (P/S) solution, 0.005 mM β-mercaptoethanol and 1% non-essential amino acids. Cells were split every two days to keep them at a density of $5 \times 10^5$ cells/ml in 37 °C, 5% $CO_2$.

## Editing LCLs with base editor plasmids
For base editing experiments, LCLs were run through Ficoll gradient and the death cells and debris were cleared. $5 \times 10^5$ LCLs were electroporated with ABEmax (750 ng) and sgRNA (250 ng) using 4D-Nucleofector™ X unit from Lonza (SF solution, DN100 (FA-55) and CM137 (FA-75)). Cells were cultured in a 24 well dish after nucleofection and transferred into a T25 flask after recovery for the long term culturing.

## Editing LCLs with in vitro transcribed mRNA
For base editing with ABEmax mRNA, $2 \times 10^5$ FA-55 or FA-75 LCLs were electroporated with 3 μg or 6 μg BE mRNA and 100 or 200 pmol of synthetic sgRNAs (Synthego), respectively. For both experiments the Lonza nucleofector was used with SF solution and the EW113 nucleofection program. Nucleofection efficiency and cell viability were assessed by flow cytometry 24 h after the nucleofection. Cells were cultured in a 96 well dish after nucleofection and transferred into a T25 flask later.

## Sanger and next generation (NGS) sequencing
Genomic DNA was extracted using QuickExtract™ DNA Extraction Solution (Lucigen) and genomic locus of the interest was amplified by using AmpliTaq Gold® 360 Master Mix (ThermoFisher Scientific). Primers for PCR and Sanger sequencing can be found in the supplemental table 3. For NGS library preparation two rounds of PCR were performed. In the first one (PCR 1), the PCR primers contained the corresponding sequence to the genomic locus and the appropriate forward and reverse Illumnia adapters sequences (supplemental table 3). PCR 2 was carried out with unique Illumina barcoding primer combinations using 15 μl of purified product from PCR1. PCR2 was purified by SPRIselect beads (Beckman). A ratio of 0.9× beads/PCR product volume was used. The resulting amplicon size and concentration was verified on the 4200 TapeStation System (Agilent) before multiplexing. For Sanger sequencing (and PCR1 for NGS) the products of the PCR were purified using MinElute columns (Qiagen) and eluted in 30 μl elution buffer (EB). -120 ng of purified PCR product was sent for Sanger economy sequencing. The forward primer was used for sequencing FA-55, while the reverse one was used for FA-75. Sanger sequencing graphs were generated using Geneious Prime 2020.2.3.

## Western blotting and MMC treatment
For MMC treatment, $2 \times 10^6$ LCLs were incubated with 1 μM MMC for 24 h before $1 \times 10^6$ cells were collected. For protein extracts, $1 \times 10^6$ LCLs were pelleted and washed in PBS. To lyse the cells, 150 μl of ice-cold RIPA buffer (Millipore) supplemented with Halt protease inhibitor

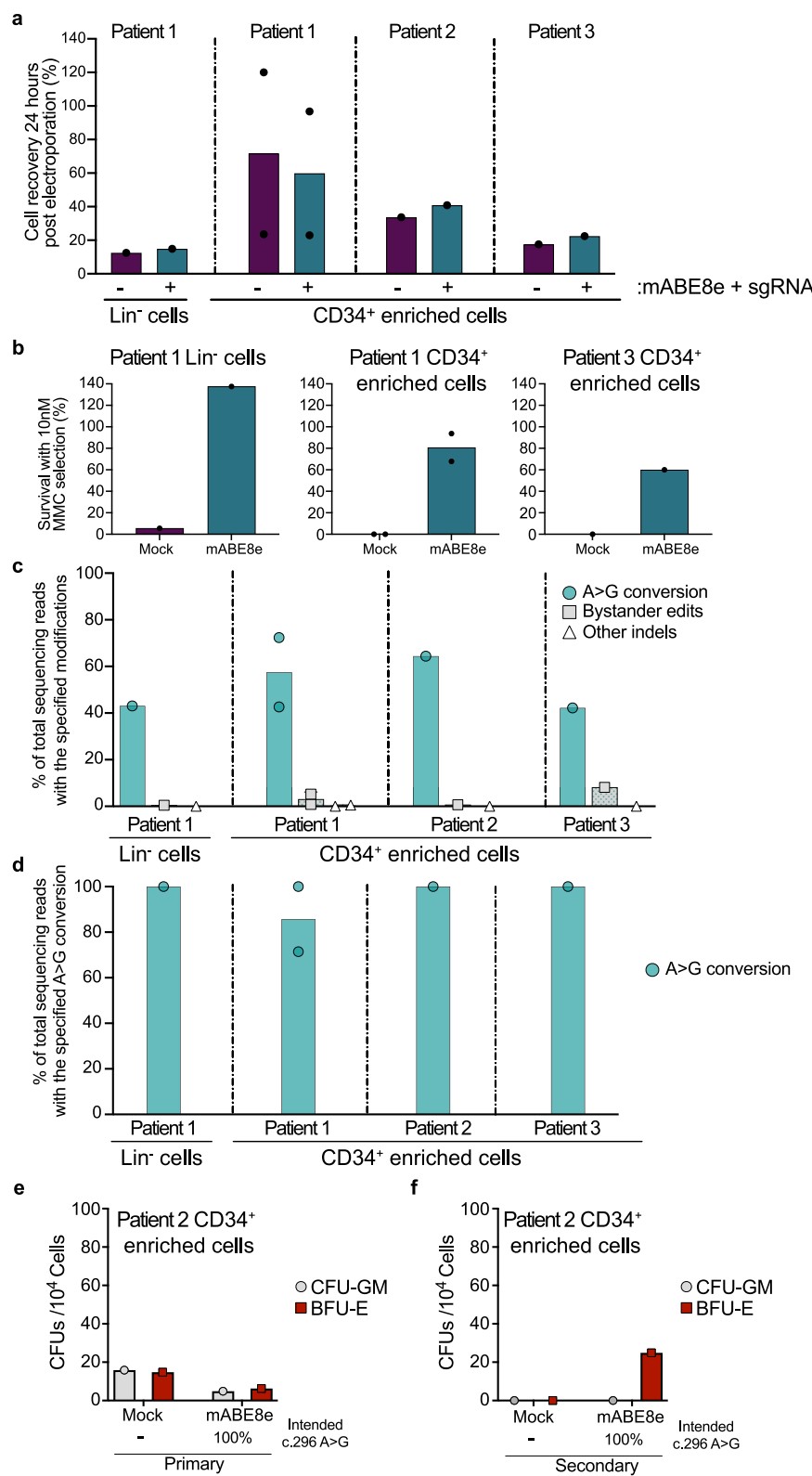

cocktail (ThermoFisher Scientific) was used. LCLs were resuspended in this lysis buffer and incubated on ice for 20 min. After centrifugation at 21,000 × *g* for 30 minutes at 4 °C the supernatant was transferred into another microcentrifuge tube. Protein concentration was measured, using Bradford Assay (VWR) and after incubation in RIPA supplemented with 1×LDS and 1×DTT for 5 minutes at 95 °C, 15 µg protein were loaded on the gel. Gel electrophoresis was run with 4–12%

polyacrylamide gels (NuPAGE) and 1×MOPS SDS running buffer (NuPAGE). Proteins were transferred using the Criterion Trans-Blot® Cell (BioRad) with a Tris-Glycine transfer buffer (25 mM Tris base, 192 mM glycine, 20% methanol (v/v); pH = 8.3). Membrane was incubated with Ponceau staining for a few minutes to confirm transfer and then blocked with 5% (w/v) non-fat dry milk in TBS-T (0.1% Tween-20) for 1 h at room temperature. Primary anti-rabbit FANCA antibody

**Fig. 6 | HSPCs from FA-A patients can be efficiently edited by ABE8e. a** Cell recovery 24 hours after ABE8e electroporation in Lin depleted cells (Lin⁻) from FA-A patient 1 and CD34⁺ enriched cells from patients FA-A 1 (two experiments were performed), patient FA-A 2 and FA-A 3. **b** Survival of hematopoietic colonies from Lin⁻ (Patient 1) and CD34⁺ enriched (Patient 1 and Patient 3) cells to MMC after mABE8e base editing. Percentage of CFUs survival is evaluated considering the number of hematopoietic colonies generated in the absence of MMC as 100%. **c** Characterization of specific A to G base conversion efficiency using adenine base editing in Lin⁻ or CD34⁺ enriched cells from four FA-A patients. Quantification of specified editing frequencies were analyzed by NGS 5 days after electroporation. **d** Sanger sequencing analysis of individual colonies from four different FA-A patients after base editing with mABE8e in the absence of MMC. The percentage of intended base conversion observed in individual colonies from mABE8e edited cells is shown. **a–d** Bars from FA-A patient 1 CD34⁺ enriched cells represent mean of two independent experiments. **e, f** Clonogenic potential of FA-A patient 2 edited cells. Primary CFUs and Secondary CFCs were obtained after replating primary hematopoietic CFUs. Number of samples analyzed $n = 1$. Source data are provided as a Source Data file.

(ab5036, Abcam), anti-rabbit FANCD2 antibody (ab221932, Abcam), anti-goat HSP60 antibody (sc-1052, Santa Cruz Biotechnology) were diluted 1:1000 in 10% milk TBS-T. HSP60 served as a loading control. The membrane was stained with antibody overnight at 4 °C and then washed three times in TBS-T before 45 min incubation with anti-rabbit secondary antibody (IRDye 800CW (926-32213) or anti-goat IRDye 800CW (926-32214), 1:5000 diluted in 10% milk TBS-T. Finally, the membrane was washed two times with TBS-T and one time with PBS before imaging with the Li-Cor's Near-InfraRed fluorescence Odyssey CLx Imaging System.

### MMC sensitivity assay
MMC sensitivity assay was performed, incubating $2.5 \times 10^5$ cells for 5 days in media with increasing concentrations of MMC (0, 3, 10, 33, 100, 333, 1000 nM). Survival was measured by flow cytometry using the forward and side scatter to gate for the life cell population. Downstream analysis was performed using FlowJo Software v10.7.1 (FlowJo, LLC). Each data point represents the mean of ≥3 biological replicates.

### NGS data analysis
Demultiplexing of the Sequencing reads was done with the MiSeq Reporter (Illumina). Sequencing reads were aligned to the genome using the bowtie2 algorithm and visualized using the Integrative genome viewer. CRISPResso2 was run in with the following settings: CRISPRessoBatch−batch_settings batch.batch−amplicon_seq -p 4 --base_edit -g -wc -10 -w 20. Corrected reads with the base edited therapeutic SNP were calculated by selecting only reads with the intended edit but no indels in the quantification window. Percentages of corrected read and uncorrected reads were plotted using GraphPad Prism 8.3.1 (GraphPad Software, Inc., San Diego, CA).

### Off-target analysis
Cas-OFFinder (http://www.rgenome.net/cas-offinder/)[39] and CRISTA (https://crista.tau.ac.il/)[40] were used to determine all possible off target sites. Cas-OFFinder was run under the following settings: mismatch number = 3 (equal or less), DNA Bulge Size = 0 and RNA Bulge Size = 0. For FA-75 sgRNA4 we analyzed the top eight of 22 potential sites, chosen by the mismatches being located outside of the seed region. Off target sites for CRISTA were selected based on the highest CRISTA score. Primer for Off target sites were designed using MRPrimerW2[52]. NGS was performed on the respective genomic sites using NGS primers listed in Supplemental Table 3. Data were analyzed by CRISPResso2 and run with the same setting as for on target base editing. For quantification of A to G conversions, all adenines or thymidines, depending on the orientation of the protospacer, were considered potential targets of the BE. Therefore, all reads which contained one or more A to G conversions in this window were scored as base edited and the sum of all reads with A to G conversions at these positions was calculated. We detected one A to G SNPs in the protospacer of FA-55 OT33 (homozygous). For the analysis, we subtracted 100% from the total edited reads, respectively. For off-target sites with A to G conversion of <0.1% in all samples, a value of 0.1% was plotted. The code used for the analysis can be found in Supplementary Fig. 10.

### Protein sequence alignment
Protein sequences for FANCA were retrieved from https://www.ncbi.nlm.nih.gov/protein and converged together. A multiple sequence alignment was created using T-Coffee (http://tcoffee.crg.cat/apps/tcoffee/do:regular) and was visualized with the help of Boxshades (https://embnet.vital-it.ch/software/BOX_form.html) and pyBoxshade. Using the "fasta_aln" result file from T-Coffee with format "other" as input and "RTF_new" as the output format.

### Hematopoietic stem and progenitor cells from healthy donors and FA patients
Human CD34⁺ cells were obtained from healthy donor umbilical cord blood (UCB) or mobilized peripheral blood samples provided by *Centro de Transfusiones de la Comunidad de Madrid* and *Hospital Infantil Universitario Niño Jesús*, respectively after informed consent was signed. Mononuclear cell fractions were purified by Ficoll-Paque PLUS (GE Healthcare) density gradient centrifugation according to manufacturer's instructions. Human CD34⁺ HSPCs were purified from the mononuclear fraction by immunoselection using the CD34 Micro-Bead Kit (MACS, Miltenyi Biotec). Magnetic-labelled cells were selected with a LS column in *QuadroMACS^TM Separator* (Miltenyi Biotec) following manufacturer's instructions. Purified hCD34⁺ were then analysed by flow cytometry to evaluate their purity in *LSRFortessa Cell Analyser* (BD) using *FlowJo Software* v10.7.1. Purities ranging from 85-98% were routinely obtained. Cells were grown in *StemSpan* (StemCell Technologies) supplemented with 1% GlutaMAX™ (Gibco), 1% P/S solution (Gibco), 100 ng/mL human stem cell factor (hSCF), human FMS-like tyrosine kinase 3 ligand (hFlt3-L), human thrombopoietin (hTPO), and 20 ng/mL human interleukin 3 (hIL3) (all obtained from EuroBiosciences) under normoxic conditions. HSPCs were pre-stimulated 24 h prior electroporation. Cryopreserved CD34 + cells were thawed and cultured under the same conditions 24 hours prior electroporation.

Lineage negative populations from FA patient 1 were obtained from apheresis aliquots by the incubation of cells with anti-hCD14-PE (BD Pharmingen), anti-hCD15-PE (Beckman Coultek), anti-hCD3-PE (Beckman Coultek), anti-hCD19-PE (Beckman Coultek), anti-hCD33-PE (eBioscience) and anti-hCD-235a-PE (BD Pharmingen) for 30 min. CD34⁺ enriched cells from patient 1 (performed in duplicate), patient 2 and patient 3 were purified from mPB by immunoselection using the CD34 Micro-Bead Kit (MACS, Miltenyi Biotec). In all cases informed consent was signed by the patients or their parents/ legal representative. Then, cells were washed and incubated with anti-PE Microbeads (Miltenyi Biotec). Lineage negative and CD34 population content was analyzed in *LSRFortessa Cell Analyser* (BD) using *FlowJo Software* v10.7.1. Cells were grown and cultured during 24 hours prior electroporation in GMP Stem Cell Grow Medium (CellGenix) supplemented with 1% GlutaMAX™ (Gibco), 1% P/S (Gibco), 100 ng/mL SCF and Flt3, 20 ng/mL TPO and IL3 (EuroBiosciences), 10 µg/mL anti-TNFα (Enbrel-Etanercept, Pfizer) and 1 mM N-acetylcysteine (Pharmazam) under hypoxic conditions (37°C, 5% of $O_2$, 5% of $CO_2$ and 95% RH).

### mRNA electroporation
Electroporation was performed using Lonza 4D Nucleofector (V4XP-3032 for 20-µl Nucleocuvette Strips or V4XP-3024 for 100-µl

Nucleocuvette Strips) according to the manufacturer's instructions. The modified synthetic sgRNA (2'-O-methyl 3' phosphorothioate modifications in the first and last three nucleotides) were purchased from Synthego and BE mRNA was obtained through in vitro transcription using mMESSAGE mMACHINE™ T3 Transcription kit (Invitrogen). A total of $2 \times 10^5$ HSPCs from healthy donor were resuspended in 20 µL P3 solution and electroporated in 20-µL Nucleocuvette wells using program EO-100 with increasing concentration of BE mRNA and sgRNA (3 µg of BE mRNA and 3.2 µg sgRNA for HD CB cells and 6 µg of BE mRNA and 6.4 µg sgRNA for HD mPB cells). For 100-µL cuvette electroporation, $1 \times 10^6$ HSPCs were resuspended in 100 µL P3 solution and electroporated using 30 µg of BE mRNA and 32 µg of sgRNA with program EO-100. FA Lineage negative cells were electroporated using similar conditions. Electroporated cells were resuspended in *StemSpan* medium (StemCell Technologies) with corresponding cytokines. Then, 24 h later, cells were used for transplant or maintained in culture for 5 days for DNA extraction and Sanger/NGS analysis to evaluate basal gene editing.

## Colony forming unit assay

Colony forming unit assays were established using 900 HD hCD34+, $7.4 \times 10^4$ FA-A hLin− or $1.8 \times 10^4$ –$1.9 \times 10^5$ FA-A CD34+ enriched cells in 3 mL of enriched methylcellulose medium (StemMACS™ HSC-CFU complete with Epo, Miltenyi Biotech). In the case of FA cells, 10 µg/mL anti-TNFα and 1mM N-acetylcysteine were added. Each mL of the triplicate was seeded in a M35 plate and incubated under normoxic (HD hCD34+ cells) or under hypoxic (FA hLin− cells) conditions. To test MMC sensitivity of hematopoietic progenitors obtained from FA-A patients, 10 nM of MMC (Sigma-Aldrich) was added to the culture. After fourteen days, colonies were counted using an inverted microscope (Nikon Diaphot, objective 4) and CFUs-GMs (granulocyte-macrophage colonies) and BFU-Es (erythroid colonies) were identified. Secondary CFUs were scored by harvesting primary colonies and plating them in methylcellulose for 14 days.

## Base editing efficiency measurement in HSPCs by NGS

Base editing frequencies were measured either from liquid cultures 5 days after electroporation or in individual hematopoietic colonies grown in methylcellulose. The *AAVS1* or *FANCA* exon 4 regions were amplified with AmpliTaq Gold 360 DNA Polymerase (Thermo Fisher Scientific) and corresponding primers using the following cycling conditions: 95 °C for 10 min; 40 cycles of 95 °C for 30 s, 60 °C for 30 s and 72 °C for 1 min; and 72 °C for 7 min. Primers used in these PCRs are listed in Supplemental Table 3. Resulting PCR products were subjected to Sanger sequencing or illumina deep sequencing. For Sanger sequencing, PCR products were sequenced using Fw primers described in Supplemental Table 3. For deep sequencing, PCR products were purified using the Zymo Research DNA Clean and Concentrator kit (#D4004), quantified using Qubit fluorometer (Thermo Fisher Scientific), and used for library construction for illumina platforms. The generated DNA fragments were sequenced by Genewiz with *Illumina MiSeq Platform*, using 250-bp paired-end sequencing reads. Frequencies of editing outcomes were quantified using CRISPResso2 software (quantification window center (-3) and size (-10); plot window size (20); base edit target A to G; batch mode).

## Mice

Non-obese diabetic (NOD) immunodeficient Cg-Prkdcscid Il2rgtm1Wjl/SzJ mice (NSG) used to test the repopulation capacity of Base edited hCD34+ cells obtained from mPB and CB. Mice were purchased from Jackson laboratories and were housed and bred at the CIEMAT Laboratory Animal Facility (registration number ES280790000183), where they were routinely screened for

pathogens in accordance with the Spanish Society for the Laboratory of Animal Science (SECAL) and the Federation of European Laboratory Animal Science Associations (FELASA).

Experiments were performed in accordance with the EU and CIEMAT guidelines upon approval as shown in *Ethical statement* section.

**Number of mice used for transplantation studies.** CB primary recipients: 7 mice (mock), 7 mice (ABEmax) and 6 mice (ABE8e). Total number of mice: 20. Sex: Female. Age: 11 weeks (Fig. 5D and Supplementary Fig. 6F).

CB secondary recipients 7 mice (mock), 7 mice (ABEmax) and 6 mice (ABE8e). Total number of mice: 20. Sex: Female. Age: 9 weeks (Fig. 5E and Supplementary Fig. 6G).

mPB primary recipients 7 mice (mock), 6 mice(ABEmax) and 6 mice (ABE8e). Total number of mice: 19. Sex: Female. Age: 11 weeks (Fig. 5G and Supplementary Fig. 6H).

## Base edited HSPCs transplantation studies in NSG mice

HD hCD34+ cells from CB or mPB were purified and pre-stimulated for 24 h for electroporation as mentioned in mRNA nucleofection section. Three groups of cells were established: electroporated cells without nuclease or sgRNA (Mock); electroporated cells with ABEmax mRNA and sgRNA (ABEmax); and electroporated cells with ABE8e mRNA and sgRNA (ABE8e). Twenty-four hours later, $3 \times 10^5$ cells per mouse were intravenously injected into immunodeficient NSG mice previously irradiated with 1.5 Gy. A CFU-assay was also conducted and the remaining cells were pelleted for DNA extraction and NGS analysis to evaluate basal gene editing. 30 and 60 days after transplantation, bone marrow samples were obtained by intra-femoral aspiration and total human engraftment was measured by flow cytometry, analyzing percentage of hCD45+ cells (anti-hCD45-FITC, BioLegend). Multilineage reconstitution was also evaluated using antibodies against hCD34 (anti-hCD34-APC, BD) for HSPCs, hCD33 (anti-hCD33-PE, eBioscience) for myeloid cells, hCD19 (anti-hCD19-Pe-Cy5, BioLegend) for B cells and hCD3 (anti-hCD3-Pe-Cy7, BioLegend) for T cells. A small aliquot of cells was used for DNA extraction and NGS analysis to evaluate the presence of gene edited cells. Mice were euthanized at 90 or 120 days post-transplantation, and bone marrow cells were obtained from hind legs. Human engraftment was evaluated by flow cytometry according to the percentage of hCD45+ cells in the different hematopoietic organs. Multilineage reconstitution was determined using antibodies against hCD34 for HSPCs, hCD33 for myeloid cells, hCD19 for B cells and hCD3 for T cells. Finally in the case of mice transplanted with CB cells, secondary recipients were transplanted using BM cells from primary recipients. Viable cells were identified by 4′, 6-diamidino-2-phenylindole (DAPI). Flow cytometry analysis were performed using a LSRFortessa Cell Analyzer and analyzed with FlowJo Software v10.7.1.

## Statistics & reproducibility

Statistical analyses shown in Fig. 5d−g and Supplementary Fig. 6f–h are conducted using GraphPad Prism software package for Windows (version 9.3.1, GraphPad Software). For the analyses of experiments in which $n \geq 5$, a Kolmogorov-Smirnov test was used to test normal distribution of the samples. In samples showing a normal distribution, an ANOVA with Turkey's multiple comparisons was performed when more than two variables were compared. Reproducibility of the results was confirmed in 6−7 mice per group as shown in Fig. 5 and Supplementary Fig. 6 and detailed in *Mice* section.

## Reporting summary

Further information on research design is available in the Nature Portfolio Reporting Summary linked to this article.

## Data availability

Sequencing data is deposited in SRA BioProject PRJNA891670. Source data are provided with this paper.

## Code availability

The code used for the analysis of off-target is available in Supplementary Fig. 10.

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

## Acknowledgements

We thank the Functional Genomics Center Zurich (FGCZ) and especially Dr. Susanne Kreutzer and Dr. Zacharias Kontarakis for their help with NGS sequencing. We thank Prof. Jordi Surrallés laboratory (San Pau Hospital, Barcelona) for kindly providing FA-55 and FA-75 LCLs and Dr. Julián Sevilla (Hospital Universitario Niño Jesús, Madrid) for providing HSPCs from HD and FA patients. We thank Lukas Villiger for sharing the ABEmax-GFP plasmid and we thank the members of the Corn Lab for helpful discussions and help with the manuscript. The laboratory of JEC has funded collaborations with Allogene. JEC is supported by the NOMIS Foundation and the Lotte and Adolf Hotz-Sprenger Stiftung. MEK is supported by the Fanconi Anemia Research Foundation. The laboratory of P.R. is supported by grants RTI2018-097125-B-I00 and PID2021-125077OB-C21 from Ministry of Science, Innovation and Universities. JEC and PR's laboratories have received funding from the European Union's Horizon 2020 research and innovation programme under the EJP RD COFUND-EJP No. 825575.

## Author contributions

S.M.S., M.E.K., P.R., J.E.C. conceived this project. S.M.S., A.C. performed the experiments from Figs. 1 to 4 and L.U. and P.R. performed the experiments Figs. 5, 6 with the help of L.G.G. in mouse experiments and NGS studies. M.E.K., J.E.C. wrote the first manuscript with the contributions from P.R. and L.U., and S.M.S. J.A.B reviewed the manuscript. All authors read and approved the final manuscript.

## Competing interests

J.E.C. is a cofounder and board member of Spotlight Therapeutics, an SAB member of Mission Therapeutics, an SAB member of Relation Therapeutics, an SAB member of Hornet Bio, an SAB member for the Joint AstraZeneca-CRUK Functional Genomics Centre, and a consultant for Cimeo Therapeutics. P.R. and J.A.B receive funding and has licensed the PGK-FANCA-WPRE* lentiviral vector to Rocket Pharmaceuticals. P.R. and J.A.B. are inventors on patents filed by CIEMAT, CIBERER, and Fundación Jiménez Díaz, and may be entitled to receive financial benefits from the licensing of such patents. The rest of the authors declare that they have no competing interests.
