## [Peer Review File · Nature Communications]

Reviewers' Comments:

Reviewer #1:

Remarks to the Author:

Summary: The authors present a novel strategy using base editing technology to correct for two common FANCA gene mutations (FA-55 and FA-75) found in patients. The article identifies adenine base-editing candidates that could correct pathogenic allele variants, and then demonstrates correction using two generations of base-editing techniques (ABEmax and ABE8e). Proof of concept studies are provided using plasmids first, followed by more clinically relevant mRNA to deliver guide RNAs and the ABE enzymes. The authors demonstrate success in base editing both in vitro, and in transplantation studies using various healthy (non FA) donor cells. In immortalized LCLs they show that crosslinker tolerance is conferred to edited FANCA deficient cells. The authors also provide evidence that few off-target sites exist, but no adverse outcomes were observed in xeno transplant recipients. Finally, the authors demonstrate restored MMC resistance in patient derived FANCA deficient bone marrow progenitor cells. Overall, the work applies base-editing as an alternative strategy to treating Fanconi Anemia.

Concerns: This work takes on a 'personalized' approach to treating FA and provides some evidence that it may have success in the clinic. However, while the editing is efficient and seems capable of restoring FA pathway function, it suffers some of the same shortcomings as existing lentiviral approaches. Ex vivo manipulation of patient hematopoietic stem cells, which are compromised in FA patients, is a primary example of one such shortcoming. Moreover, for FA cells to undergo nucleofection, which may exaggerate apoptotic attrition during ex vivo culture. From the data presented in Figure 6, it appears there may be a significant impact on cell viability post-nucleofection in FA cells (much fewer colonies in nucleofected groups) which might preclude any therapeutic effect and make a significant impact in the clinic. Data from a single patient limited only to colony formation in MMC limits any argument in favor of the efficacy of this approach in correcting FA HSPCs.

Major:

- The authors point out the bias associated with their methods for off-target editing identification. The use of a more comprehensive approach, utilizing multiple prediction algorithms to off-target site identification and the quantification of off-target activity would provide a more reliable assessment of the safety of this approach.
- Additional functional assays in FA compromised cells are warranted. The authors offer an N of 1 and a single type of assay due to the scarcity of patient-derived FA cells. Increased sample number and a more diverse panel of assays (e.g. transplantation/serial transplantation, serial replating in CFU assay, or assessments of FA HSPC 'stemness' by cell progeny distributions) would greatly increase the impact of the work and provide a more comprehensive analysis of the clinical relevance of this approach in the context of FA.
- A more thorough assessment of cell viability/recovery after nucleofection and base-editing, especially in the context of FA would be good to see. The supplemental figure provides some insight toward recovery after nucleofection, but these results are in healthy donor cells only.
-

Minor:

- -In general, the article needs additional proofreading. Some typos and errors found in the main text:
 - -Line 48 guardian->guard
 - -Line 64 facilitate to "surgically" change -> facilitate "surgically" changing
 - -Line 92 "an"-> "a"
 - -Line 137 double stranded-> double-stranded
 - -145 "mRNA-based"
 - -Line 257 refers to supp. Fig. 'H', which does not exist.
 - -Lines 310-312 Run-on sentence needs to be edited
 - -Line 326 active -> activities

- -Line 498 colum -> column
- -Line 502/532 StemSpam->StemSpan

Reviewer #2:

Remarks to the Author:

In this report, Siegner and colleagues investigate adenine base editing as an approach to correct point mutations underlying Fanconi anemia. There are many different affected genes and many mutations per gene in FA. This report focuses on 2 point mutations in FANCA (one each from the FA-55 and FA-57 donors) that are favorable candidates for correction by adenine base editing.

The results are clear that adenine base editors, when delivered to cells as ABE mRNA and synthetic sgRNA, can efficiently convert A>G and correct MMC sensitivity of FA cells. Moreover the ABE method (particularly with ABE8e mRNA) was efficient and non-toxic in HSPCs which could demonstrate engraftment of edited cells for 90 days after infusion.

Although the results are not extremely surprising given other recent reports of efficient CBE/ABE in HSPCs (e.g. Zeng et al, Nat Med 2020, Newby et al Nature 2021, Chu et al CRISPR J 2021, Knipping et al, Mol Ther 2022) albeit for other target loci and the impact on FA may be limited to a subset of mutations amenable to the ABE8e system (in terms of mutation type, available PAM, and potential bystander edits), this report demonstrates that ABE is highly promising to achieve ex vivo therapeutic base editing where the target site is permissive, and that this could be especially favorable for a disease like FA where the corrected cells gain a substantial advantage over the uncorrected cells.

Some specific comments/questions:

1. What is the prior evidence that the Gln>Trp missense substitution in FANCA would be tolerated for the FA-55 c.295 C>T modification strategy? In terms of sequence conservation there is some variability at this sequence (Fig 1A) but none of the other orthologous sequences shown have Trp in this position so it is not totally apparent that the missense mutation would be fully functional.
2. For the experiments in Fig. 1, it is hard to disentangle how much the increase in editing over time may reflect positive selection due to FANCA functional correction as compared to long duration of expression of the mRNA/sgRNA that could continue to edit after day 5. It would be helpful to show similar experiment for an edit without selection such as the AAVS1 edit explored later, to distinguish these effects.
3. For Fig 2C, why is there less basal FANCD2-Ub compared to HD FA-55? Is this because the missense mutation doesn't completely rescue FANCA level and/or function in face of endogenous DNA damage triggers (even if it rescues survival after MMC)?
4. Likewise in Fig3E, for FA-75, it is not clear from the Western blot that FANCD2-Ub is restored after editing as compared to HD. Does this indicate only partial recovery of molecular function, perhaps due to unexpected effects of bystander mutations (even if bystanders are synonymous, might they have a regulatory impact)?
5. For Fig 5, I suggest to show the protospacer and PAM to make it clear what position with respect to protospacer (and thus base editing window) is the target A. It is not obvious where are the bystander edits, since there are no nearby A's to the target. Suggest to also show representative table of alleles and their frequencies, so that the nature of the bystanders and indels are more apparent.
6. Showing the edited alleles and frequencies in a table, even if in a supplemental figure, would be helpful for all the editing scenarios in this report (all the tested ABEs at all the target loci).
7. For Fig 6, why only homozygous clones found even without MMC? Was the overall editing higher in this experiment as compared to other experiments with same editors? Or is the selective pressure for correction of both alleles higher in methylcellulose as compared to liquid culture conditions? It would be helpful to report the editing in liquid cultured cells with this donor.
8. Was there an interferon response of HSPCs following mRNA/sgRNA electroporation?
9. The authors state that unbiased identification of genomic and transcriptomic off-targets are an important next step. I suggest the authors provide more details about what kind of study design they anticipate, and power to perform unbiased genomic off-target analysis, including for gRNA-

independent deaminase effects.

10. Why is editing at 5 days ~30% in Fig 3b but ~50% in Fig 1C (for FA-55) with seemingly same conditions?

Minor:

- There are some issues with the grammar and syntax throughout, e.g. line 59, so suggest to carefully copy edit the entire manuscript.
- If space is limited, it would seem that some of the figures could be combined or some could be moved to supplement.

Reviewer #3:

Remarks to the Author:

Fanconi anemia (FA) is a prototypical hereditary disorder that is deficient in DNA repair, stem cell maintenance, and genome stability. The patient-derived hematopoietic stem cells might be corrected by lentiviral transduction, however, a more desirable way for gene therapy is the correction of the specific causative FA gene mutations. In this manuscript, the authors utilized two base editing systems and showed that they could correct the two FANCA mutations even in FA patient-derived hematopoietic stem/progenitor cells with excellent efficiency. As expected, the gene correction restores functional defects in the FA pathway such as MMC sensitivity.

In essence, this paper described the application of preexisting base editing technology to an important genetic disorder. The results were properly described with high standards. I have only a few comments.

1. P.2, line 40. The number of the so far identified FA gene is 22, not 23 as described in the cited review (Ref 8)
2. The corrected FANCA mutations are not necessarily "the most prevalent" (p.3, line 75). That depends on the ethnic group, and they should specify in which population they are most prevalent.

Reviewer #1 (Remarks to the Author):

Summary: The authors present a novel strategy using base editing technology to correct for two common FANCA gene mutations (FA-55 and FA-75) found in patients. The article identifies adenine base-editing candidates that could correct pathogenic allele variants, and then demonstrates correction using two generations of base-editing techniques (ABEmax and ABE8e). Proof of concept studies are provided using plasmids first, followed by more clinically relevant mRNA to deliver guide RNAs and the ABE enzymes. The authors demonstrate success in base editing both in vitro, and in transplantation studies using various healthy (non FA) donor cells. In immortalized LCLs they show that crosslinker tolerance is conferred to edited FANCA deficient cells. The authors also provide evidence that few off-target sites exist, but no adverse outcomes were observed in xeno transplant recipients. Finally, the authors demonstrate restored MMC resistance in patient derived FANCA deficient bone marrow progenitor cells. Overall, the work applies base-editing as an alternative strategy to treating Fanconi Anemia.

Concerns: This work takes on a 'personalized' approach to treating FA and provides some evidence that it may have success in the clinic. However, while the editing is efficient and seems capable of restoring FA pathway function, it suffers some of the same shortcomings as existing lentiviral approaches. Ex vivo manipulation of patient hematopoietic stem cells, which are compromised in FA patients, is a primary example of one such shortcoming. Moreover, for FA cells to undergo nucleofection, which may exaggerate apoptotic attrition during ex vivo culture. From the data presented in Figure 6, it appears there may be a significant impact on cell viability post-nucleofection in FA cells (much fewer colonies in nucleofected groups) which might preclude any therapeutic effect and make a significant impact in the clinic. Data from a single patient limited only to colony formation in MMC limits any argument in favor of the efficacy of this approach in correcting FA HSPCs.

We thank the referee for this important context. Our goal here was to explore alternative ex vivo approaches to treat the bone marrow failure associated with FA, and not to develop an in vivo therapy. There is currently no prevalent approach to deliver genome editing reagents to the bone marrow in situ. But we note that the base editing strategy described here could be combined with rapid developments in LNP technology. This has been added to the Discussion. With respect to Figure 6, we have now added a large amount of data regarding cell and CFUs recovery after electroporation. We also included transplant from HD CD34+ cells (Figure 5) and secondary CFU data from FA patients (Figure 6F). These all highlight that HSPCs from HD and FA cells tolerate base editing well.

Major:

- The authors point out the bias associated with their methods for off-target editing identification. The use of a more comprehensive approach, utilizing multiple prediction algorithms to off-target site identification and the quantification of off-target activity would provide a more reliable assessment of the safety of this approach.

We thank the reviewer for his/her advice and following his/her direction, we used the CRISTA and have now analyzed 40 potential off target sites for guide RNA (sgRNA1) targeting FA-55 mutation and 29 potential OT sites for guide RNA (sgRNA4) targeting FA-75 mutation. We

performed amplicon seq for each site and analyzed the results for the potential off targets (Figure 4, Supplementary Fig. 5). We did not see any off-target activity for sgRNA 1. However, we detected low levels of off-targets for sgRNA4 when edited with ABE8e at OT11, OT16 and OT27 (Supplementary Fig. 5) which are located in an intergenic region and intron of the *TRIO* and *ZNF267* gene, respectively. We also performed amplicon sequencing in FA HSPCs from two patients edited with mABE8e and sgRNA1 (Supplementary Fig. 8). Confirming the results found in FA-55 LCLs, we could not detect any elevated A to G conversion at OT1-8 and OT38 in the edited samples. Our results emphasize that ABE8e activity should be carefully analyzed for off targets but that the presence of OTs depends on the gRNA. We note that OTs themselves are not a barrier to a potential therapy. But their location and functional impact must be well understood for example in GLP tox studies.

- Additional functional assays in FA compromised cells are warranted. The authors offer an N of 1 and a single type of assay due to the scarcity of patient-derived FA cells. Increased sample number and a more diverse panel of assays (e.g. transplantation/serial transplantation, serial replating in CFU assay, or assessments of FA HSPC 'stemness' by cell progeny distributions) would greatly increase the impact of the work and provide a more comprehensive analysis of the clinical relevance of this approach in the context of FA.

We thank the reviewer for her/his suggestion.

In this version of the manuscript, we have included new results from a total of four samples from three FA-A patients' CD34 enriched cells (Figure 6).

Results obtained in the four CD34+ enriched cells from three independent patients confirmed the high editing efficiency by ABE8 and the maintenance of clonogenic capacity after BE (Supplementary Figure 7). As suggested by the reviewer we also performed serial replating CFU assay in CD34+ enriched cells from one of the patients and we confirmed that only cells edited show replating capacity (Figure 6F).

We also found reversion of MMC sensitivity in base edited CD34 enriched cells in samples from patient 02002 and 02005 (Figure 6B).

Furthermore, in four samples the proliferative advantage of edited cells was observed from day 5 until day 14 in CFCs, which confirms the correction of the FA phenotype by the therapeutic SNP (Figure 6D).

As previously mentioned, the number of cells obtained was too low to perform *in vitro* and *in vivo* studies.

- A more thorough assessment of cell viability/recovery after nucleofection and base-editing, especially in the context of FA would be good to see. The supplemental figure provides some insight toward recovery after nucleofection, but these results are in healthy donor cells only.

We thank the reviewer for highlighting this important point.

We have included in Figure 6A the survival after electroporation in HSPCs from FA patient mock electroporated or electroporated with BE. Similar survival was observed in both cases. Unfortunately, similar studies could not be performed in FA patient samples that were not electroporated (due to the low number of cells available). We note that frozen aliquots of FA samples have been used in these experiments and this also explains the reduction in cell survival observed in these cells, as we previously described (Jacome et al., 2009).

In addition to experiments suggested by the reviewer we have also included results from secondary recipients in Figure 5E and F and supplementary figure 6G.

Minor:

- -In general, the article needs additional proofreading. Some typos and errors found in the main text:

- -Line 48 guardian->guard

We thank the reviewer spotting the mistake and we corrected it on the text accordingly.

- -Line 64 facilitate to “surgically” change -> facilitate “surgically” changing

We thank the reviewer spotting the mistake and we corrected it on the text accordingly

- -Line 92 “an”-> “a”

We thank the reviewer spotting the mistake and we corrected it on the text accordingly

- -Line 137 double stranded-> double-stranded

We thank the reviewer spotting the mistake and we corrected it on the text accordingly

- -145 “mRNA-based”

We thank the reviewer spotting the mistake and we corrected it on the text accordingly

- -Line 257 refers to supp. Fig. ‘H’, which does not exist.

We thank the reviewer spotting the mistake and we corrected it on the text accordingly

- -Lines 310-312 Run-on sentence needs to be edited

We thank the reviewer spotting the mistake and we corrected it on the text accordingly

- -Line 326 active -> activities

We thank the reviewer spotting the mistake and we corrected it on the text accordingly

- -Line 498 colum -> column

We thank the reviewer spotting the mistake and we corrected it on the text accordingly

- -Line 502/532 StemSpam->StemSpan

We thank the reviewer spotting the mistake and we corrected it on the text accordingly

Reviewer #2 (Remarks to the Author):

In this report, Siegner and colleagues investigate adenine base editing as an approach to correct point mutations underlying Fanconi anemia. There are many different affected genes and many mutations per gene in FA. This report focuses on 2 point mutations in FANCA (one each from the FA-55 and FA-57 donors) that are favorable candidates for correction by adenine base editing.

The results are clear that adenine base editors, when delivered to cells as ABE mRNA and synthetic sgRNA, can efficiently convert A>G and correct MMC sensitivity of FA cells. Moreover the ABE method (particularly with ABE8e mRNA) was efficient and non-toxic in HSPCs which could demonstrate engraftment of edited cells for 90 days after infusion.

Although the results are not extremely surprising given other recent reports of efficient CBE/ABE in HSPCs (e.g. Zeng et al, Nat Med 2020, Newby et al Nature 2021, Chu et al CRISPR J 2021, Knipping et al, Mol Ther 2022) albeit for other target loci and the impact on FA may be limited to a subset of mutations amenable to the ABE8e system (in terms of mutation type, available PAM, and potential bystander edits), this report demonstrates that ABE is highly promising to achieve ex vivo therapeutic base editing where the target site is permissive, and that this could be especially favorable for a disease like FA where the corrected cells gain a substantial advantage over the uncorrected cells.

Some specific comments/questions:

1. What is the prior evidence that the Gln>Trp missense substitution in FANCA would be tolerated for the FA-55 c.295 C>T modification strategy? In terms of sequence conservation there is some variability at this sequence (Fig 1A) but none of the other orthologous sequences shown have Trp in this position so it is not totally apparent that the missense mutation would be fully functional.

We thank the reviewer for this comment.

Overall phenotypic characterization of both edited patient derived LCLs and the patient cells showed functionality of the missense mutation. However, this comment made us re-analyze the sequence alignments with broader species diversity and we found that in the northern sea otter and eurasian otter FANCA has Trp at this exact position, we included our analysis in the Supplementary Fig. 2.

2. For the experiments in Fig. 1, it is hard to disentangle how much the increase in editing over time may reflect positive selection due to FANCA functional correction as compared to long duration of expression of the mRNA/sgRNA that could continue to edit after day 5. It would be helpful to show similar experiment for an edit without selection such as the AAVS1 edit explored later, to distinguish these effects.

We thank the reviewer for this comment.

This is an excellent point. We disentangled the effects of FANCA correction vs. expression duration by examining the prevalence of the correction mutation vs. a silent SNP also induced by the FA-75 targeting sgRNA. We found that NGS reads containing the corrected base increase over time but reads containing the silent bystander only do not (Supplementary

Fig.3). While we cannot completely rule out the chance of retargeting of the sequences with the silent bystander, this suggests no additional editing later than day 5 after nucleofection. Moreover, Jiang et al 2020, Nat Comm (in supplemental figure 1A) performed western blots to track Cas9 and base editor expression from mRNA and found no remaining expressed protein 48 hours after nucleofection. These data strongly suggest that the increase in editing is due to functional correction and not extended expression.

3. For Fig 2C, why is there less basal FANCD2-Ub compared to HD FA-55? Is this because the missense mutation doesn't completely rescue FANCA level and/or function in face of endogenous DNA damage triggers (even if it rescues survival after MMC)?

4. Likewise in Fig3E, for FA-75, it is not clear from the Western blot that FANCD2-Ub is restored after editing as compared to HD. Does this indicate only partial recovery of molecular function, perhaps due to unexpected effects of bystander mutations (even if bystanders are synonymous, might they have a regulatory impact)?

We thank the reviewer for these comments.

Since we performed these experiments with edited pools, it is possible that the FA-55 population still contained mutated cells or heterozygous cells reflecting lower FANCD2-Ub levels. As can be seen in Figure 1C, FA-55 edited alleles were stabilized around 60% after 30 days post nucleofection, which is when we performed the western blot analysis. Given that FA-55 edited patient and LCLs show proliferative advantage in our experiments, we believe that the missense mutation restores FANCA function to a great extent.

We apologize if the previous Western blots were unclear.

We have repeated the experiment for Figure 3 for FA-55 and FA-75 cells and performed the western blot for both cell lines. These clearly indicate the restoration of FANCD2-Ub in FA-75 cells. Also note our analysis above, showing that the bystander mutations do not accumulate over time, but corrected alleles do.

5. For Fig 5, I suggest to show the protospacer and PAM to make it clear what position with respect to protospacer (and thus base editing window) is the target A. It is not obvious where are the bystander edits, since there are no nearby A's to the target. Suggest to also show representative table of alleles and their frequencies, so that the nature of the bystanders and indels are more apparent.

We thank the reviewer for this suggestion.

We have updated the Figure 5A according to the reviewer's advice and we also included a raw data in supplemental figure 9 to highlight the bystander edit. The bystander edit is not in canonical editing window of ABE but since our data points the presence of such edit albeit very low percent, we included this in our data. We hope this will clarify the type of bystander mutations to the reader.

6. Showing the edited alleles and frequencies in a table, even if in a supplemental figure, would be helpful for all the editing scenarios in this report (all the tested ABEs at all the target loci).

We thank the reviewer for this suggestion.

Now we have included Supplementary Figure 4 and 9 to show the exact editing events in all the different editing scenarios.

7. For Fig 6, why only homozygous clones found even without MMC? Was the overall editing higher in this experiment as compared to other experiments with same editors? Or is the selective pressure for correction of both alleles higher in methylcellulose as compared to liquid culture conditions? It would be helpful to report the editing in liquid cultured cells with this donor.

We thank the reviewer for this comment.

We also observed similar % of homozygous CFCs when ABE8e used in HD CB34+ cells (Supplementary Fig. 6D and E). For this reason we think that this level of homozygosis is associated to the high editing percentage obtained using ABE8e. Gene editing in liquid culture has been now included in Fig.6C.

8. Was there an interferon response of HSPCs following mRNA/sgRNA electroporation?

We thank the reviewer for this comment.

Although interferon response has not been measured in our study, previous studies have associated the use of phosphorylated IVT sgRNA. This effect is ablated by the use of chemically modified sgRNAs (Wienert et al., 2018 & Kim et al., 2018).

9. The authors state that unbiased identification of genomic and transcriptomic off-targets are an important next step. I suggest the authors provide more details about what kind of study design they anticipate, and power to perform unbiased genomic off-target analysis, including for gRNA-independent deaminase effects.

We thank the reviewer for this comment.

We have inserted detailed experimental suggestions in the discussion section, suggesting whole genome sequencing and RNA sequencing before and after base edit experiments.

10. Why is editing at 5 days ~30% in Fig 3b but ~50% in Fig 1C (for FA-55) with seemingly same conditions?

We thank the reviewer for this comment.

Though we try to keep the same conditions for our electroporation experiments, variability among different batches of mRNA and the proliferation status of the cells can lead to this variability.

Minor:

- There are some issues with the grammar and syntax throughout, e.g. line 59, so suggest to carefully copy edit the entire manuscript.

We thank the reviewer for spotting these mistakes and we corrected grammar and syntax in the text accordingly.

- If space is limited, it would seem that some of the figures could be combined or some could be moved to supplement.

Reviewer #3 (Remarks to the Author):

Fanconi anemia (FA) is a prototypical hereditary disorder that is deficient in DNA repair, stem cell maintenance, and genome stability. The patient-derived hematopoietic stem cells might be corrected by lentiviral transduction, however, a more desirable way for gene therapy is the correction of the specific causative FA gene mutations. In this manuscript, the authors utilized two base editing systems and showed that they could correct the two FANCA mutations even in FA patient-derived hematopoietic stem/progenitor cells with excellent efficiency. As expected, the gene correction restores functional defects in the FA pathway such as MMC sensitivity.

In essence, this paper described the application of preexisting base editing technology to an important genetic disorder. The results were properly described with high standards. I have only a few comments.

We thank the referee for the enthusiasm for our manuscript and suggestions.

1. P.2, line 40. The number of the so far identified FA gene is 22, not 23 as described in the cited review (Ref 8)

We thank the reviewer for spotting the mistake and we corrected it in the text accordingly.

2. The corrected FANCA mutations are not necessarily “the most prevalent” (p.3, line 75). That depends on the ethnic group, and they should specify in which population they are most prevalent.

We thank the reviewer for spotting the inconsistency and we specified it in the text.

Reviewers' Comments:

Reviewer #1:

Remarks to the Author:

The reviewer appreciates the responsive revision by the authors!

Reviewer #2:

Remarks to the Author:

Overall the revision was responsive to my concerns and the manuscript appears improved.

The sentence on lines 380/381 could be revised for clarity ("stereotyped unedited edits" is confusing). I think the point is in addition to off-target edits at homologous sequences, the deaminase may cause non-stereotyped edits to DNA or RNA.

Point by Point Response to Reviewer Comments on the final submission

Reviewer #1:

The reviewer appreciates the responsive revision by the authors!

We thanked the reviewer for the great comment.

Reviewer #2:

Overall the revision was responsive to my concerns and the manuscript appears improved.

The sentence on lines 380/381 could be revised for clarity (“stereotyped unedited edits” is confusing). I think the point is in addition to off-target edits at homologous sequences, the deaminase may cause non-stereotyped edits to DNA or RNA.

We thanked the reviewer for the great comment and we fixed the sentence for better clarification.

Point by Point Response to Reviewer Comments on the revised submission

Reviewer #1 (Remarks to the Author):

Summary: The authors present a novel strategy using base editing technology to correct for two common FANCA gene mutations (FA-55 and FA-75) found in patients. The article identifies adenine base-editing candidates that could correct pathogenic allele variants, and then demonstrates correction using two generations of base-editing techniques (ABEmax and ABE8e). Proof of concept studies are provided using plasmids first, followed by more clinically relevant mRNA to deliver guide RNAs and the ABE enzymes. The authors demonstrate success in base editing both in vitro, and in transplantation studies using various healthy (non FA) donor cells. In immortalized LCLs they show that crosslinker tolerance is conferred to edited FANCA deficient cells. The authors also provide evidence that few off-target sites exist, but no adverse outcomes were observed in xeno transplant recipients. Finally, the authors demonstrate restored MMC resistance in patient derived FANCA deficient bone marrow progenitor cells. Overall, the work applies base-editing as an alternative strategy to treating Fanconi Anemia.

Concerns: This work takes on a ‘personalized’ approach to treating FA and provides some evidence that it may have success in the clinic. However, while the editing is efficient and

seems capable of restoring FA pathway function, it suffers some of the same shortcomings as existing lentiviral approaches. Ex vivo manipulation of patient hematopoietic stem cells, which are compromised in FA patients, is a primary example of one such shortcoming. Moreover, for FA cells to undergo nucleofection, which may exaggerate apoptotic attrition during ex vivo culture. From the data presented in Figure 6, it appears there may be a significant impact on cell viability post-nucleofection in FA cells (much fewer colonies in nucleofected groups) which might preclude any therapeutic effect and make a significant impact in the clinic. Data from a single patient limited only to colony formation in MMC limits any argument in favor of the efficacy of this approach in correcting FA HSPCs.

We thank the referee for this important context. Our goal here was to explore alternative ex vivo approaches to treat the bone marrow failure associated with FA, and not to develop an in vivo therapy. There is currently no prevalent approach to deliver genome editing reagents to the bone marrow in situ. But we note that the base editing strategy described here could be combined with rapid developments in LNP technology. This has been added to the Discussion. With respect to Figure 6, we have now added a large amount of data regarding cell and CFUs recovery after electroporation. We also included transplant from HD CD34+ cells (Figure 5) and secondary CFU data from FA patients (Figure 6F). These all highlight that HSPCs from HD and FA cells tolerate base editing well.

Major:

- The authors point out the bias associated with their methods for off-target editing identification. The use of a more comprehensive approach, utilizing multiple prediction algorithms to off-target site identification and the quantification of off-target activity would provide a more reliable assessment of the safety of this approach.

We thank the reviewer for his/her advice and following his/her direction, we used the CRISTA and have now analyzed 40 potential off target sites for guide RNA (sgRNA1) targeting FA-55 mutation and 29 potential OT sites for guide RNA (sgRNA4) targeting FA-75 mutation. We performed amplicon seq for each site and analyzed the results for the potential off targets (Figure 4, Supplementary Fig. 5). We did not see any off-target activity for sgRNA 1. However, we detected low levels of off-targets for sgRNA4 when edited with ABE8e at OT11, OT16 and OT27 (Supplementary Fig. 5) which are located in an intergenic region and intron of the *TRIO* and *ZNF267* gene, respectively. We also performed amplicon sequencing in FA HSPCs from two patients edited with mABE8e and sgRNA1 (Supplementary Fig. 8). Confirming the results found in FA-55 LCLs, we could not detect any elevated A to G conversion at OT1-8 and OT38 in the edited samples. Our results emphasize that ABE8e activity should be carefully analyzed for off targets but that the presence of OTs depends on the gRNA. We note that OTs themselves are not a barrier to a potential therapy. But their location and functional impact must be well understood for example in GLP tox studies.

- Additional functional assays in FA compromised cells are warranted. The authors offer an N of 1 and a single type of assay due to the scarcity of patient-derived FA cells. Increased sample number and a more diverse panel of assays (e.g. transplantation/serial transplantation, serial replating in CFU assay, or assessments of FA HSPC 'stemness' by cell progeny distributions) would greatly increase the impact of the work and provide a more comprehensive analysis of the clinical relevance of this approach in the context of FA.

We thank the reviewer for her/his suggestion.

In this version of the manuscript, we have included new results from a total of four samples from three FA-A patients' CD34 enriched cells (Figure 6).

Results obtained in the four CD34+ enriched cells from three independent patients confirmed the high editing efficiency by ABE8 and the maintenance of clonogenic capacity after BE (Supplementary Figure 7). As suggested by the reviewer we also performed serial replating CFU assay in CD34+ enriched cells from one of the patients and we confirmed that only cells edited show replating capacity (Figure 6F).

We also found reversion of MMC sensitivity in base edited CD34 enriched cells in samples from patient 02002 and 02005 (Figure 6B).

Furthermore, in four samples the proliferative advantage of edited cells was observed from day 5 until day 14 in CFCs, which confirms the correction of the FA phenotype by the therapeutic SNP (Figure 6D).

As previously mentioned, the number of cells obtained was too low to perform *in vitro* and *in vivo* studies.

- A more thorough assessment of cell viability/recovery after nucleofection and base-editing, especially in the context of FA would be good to see. The supplemental figure provides some insight toward recovery after nucleofection, but these results are in healthy donor cells only.

We thank the reviewer for highlighting this important point.

We have included in Figure 6A the survival after electroporation in HSPCs from FA patient mock electroporated or electroporated with BE. Similar survival was observed in both cases. Unfortunately, similar studies could not be performed in FA patient samples that were not electroporated (due to the low number of cells available). We note that frozen aliquots of FA samples have been used in these experiments and this also explains the reduction in cell survival observed in these cells, as we previously described (Jacome et al., 2009).

In addition to experiments suggested by the reviewer we have also included results from secondary recipients in Figure 5E and F and supplementary figure 6G.

Minor:

- -In general, the article needs additional proofreading. Some typos and errors found in the main text:

- -Line 48 guardian->guard

We thank the reviewer spotting the mistake and we corrected it on the text accordingly.

- -Line 64 facilitate to "surgically" change -> facilitate "surgically" changing

We thank the reviewer spotting the mistake and we corrected it on the text accordingly

- -Line 92 "an"-> "a"

We thank the reviewer spotting the mistake and we corrected it on the text accordingly
- -Line 137 double stranded-> double-stranded

We thank the reviewer spotting the mistake and we corrected it on the text accordingly
- -145 “mRNA-based”

We thank the reviewer spotting the mistake and we corrected it on the text accordingly
- -Line 257 refers to supp. Fig. ‘H’, which does not exist.

We thank the reviewer spotting the mistake and we corrected it on the text accordingly
- -Lines 310-312 Run-on sentence needs to be edited

We thank the reviewer spotting the mistake and we corrected it on the text accordingly
- -Line 326 active -> activities

We thank the reviewer spotting the mistake and we corrected it on the text accordingly
- -Line 498 colum -> column

We thank the reviewer spotting the mistake and we corrected it on the text accordingly

- -Line 502/532 StemSpam->StemSpan

We thank the reviewer spotting the mistake and we corrected it on the text accordingly

Reviewer #2 (Remarks to the Author):

In this report, Siegner and colleagues investigate adenine base editing as an approach to correct point mutations underlying Fanconi anemia. There are many different affected genes and many mutations per gene in FA. This report focuses on 2 point mutations in FANCA (one each from the FA-55 and FA-57 donors) that are favorable candidates for correction by adenine base editing.

The results are clear that adenine base editors, when delivered to cells as ABE mRNA and synthetic sgRNA, can efficiently convert A>G and correct MMC sensitivity of FA cells. Moreover the ABE method (particularly with ABE8e mRNA) was efficient and non-toxic in HSPCs which could demonstrate engraftment of edited cells for 90 days after infusion.

Although the results are not extremely surprising given other recent reports of efficient CBE/ABE in HSPCs (e.g. Zeng et al, Nat Med 2020, Newby et al Nature 2021, Chu et al CRISPR J 2021, Knipping et al, Mol Ther 2022) albeit for other target loci and the impact on FA may be limited to a subset of mutations amenable to the ABE8e system (in terms of mutation type, available PAM, and potential bystander edits), this report demonstrates that ABE is highly promising to achieve ex vivo therapeutic base editing where the target site is permissive, and that this could be especially favorable for a disease like FA where the corrected cells gain a substantial advantage over the uncorrected cells.

Some specific comments/questions:

1. What is the prior evidence that the Gln>Trp missense substitution in FANCA would be tolerated for the FA-55 c.295 C>T modification strategy? In terms of sequence conservation there is some variability at this sequence (Fig 1A) but none of the other orthologous sequences

shown have Trp in this position so it is not totally apparent that the missense mutation would be fully functional.

We thank the reviewer for this comment.

Overall phenotypic characterization of both edited patient derived LCLs and the patient cells showed functionality of the missense mutation. However, this comment made us re-analyze the sequence alignments with broader species diversity and we found that in the northern sea otter and eurasian otter FANCA has Trp at this exact position, we included our analysis in the Supplementary Fig. 2.

2. For the experiments in Fig. 1, it is hard to disentangle how much the increase in editing over time may reflect positive selection due to FANCA functional correction as compared to long duration of expression of the mRNA/sgRNA that could continue to edit after day 5. It would be helpful to show similar experiment for an edit without selection such as the AAVS1 edit explored later, to distinguish these effects.

We thank the reviewer for this comment.

This is an excellent point. We disentangled the effects of FANCA correction vs. expression duration by examining the prevalence of the correction mutation vs. a silent SNP also induced by the FA-75 targeting sgRNA. We found that NGS reads containing the corrected base increase over time but reads containing the silent bystander only do not (Supplementary Fig.3). While we cannot completely rule out the chance of retargeting of the sequences with the silent bystander, this suggests no additional editing later than day 5 after nucleofection. Moreover, Jiang et al 2020, Nat Comm (in supplemental figure 1A) performed western blots to track Cas9 and base editor expression from mRNA and found no remaining expressed protein 48 hours after nucleofection. These data strongly suggest that the increase in editing is due to functional correction and not extended expression.

3. For Fig 2C, why is there less basal FANCD2-Ub compared to HD FA-55? Is this because the missense mutation doesn't completely rescue FANCA level and/or function in face of endogenous DNA damage triggers (even if it rescues survival after MMC)?

4. Likewise in Fig3E, for FA-75, it is not clear from the Western blot that FANCD2-Ub is restored after editing as compared to HD. Does this indicate only partial recovery of molecular function, perhaps due to unexpected effects of bystander mutations (even if bystanders are synonymous, might they have a regulatory impact)?

We thank the reviewer for these comments.

Since we performed these experiments with edited pools, it is possible that the FA-55 population still contained mutated cells or heterozygous cells reflecting lower FANCD2-Ub levels. As can be seen in Figure 1C, FA-55 edited alleles were stabilized around 60% after 30 days post nucleofection, which is when we performed the western blot analysis. Given that FA-55 edited patient and LCLs show proliferative advantage in our experiments, we believe that the missense mutation restores FANCA function to a great extent.

We apologize if the previous Western blots were unclear.

We have repeated the experiment for Figure 3 for FA-55 and FA-75 cells and performed the western blot for both cell lines. These clearly indicate the restoration of FANCD2-Ub in FA-75 cells. Also note our analysis above, showing that the bystander mutations do not accumulate over time, but corrected alleles do.

5. For Fig 5, I suggest to show the protospacer and PAM to make it clear what position with respect to protospacer (and thus base editing window) is the target A. It is not obvious where are the bystander edits, since there are no nearby A's to the target. Suggest to also show representative table of alleles and their frequencies, so that the nature of the bystanders and indels are more apparent.

We thank the reviewer for this suggestion.

We have updated the Figure 5A according to the reviewer's advice and we also included a raw data in supplemental figure 9 to highlight the bystander edit. The bystander edit is not in canonical editing window of ABE but since our data points the presence of such edit albeit very low percent, we included this in our data. We hope this will clarify the type of bystander mutations to the reader.

6. Showing the edited alleles and frequencies in a table, even if in a supplemental figure, would be helpful for all the editing scenarios in this report (all the tested ABEs at all the target loci).

We thank the reviewer for this suggestion.

Now we have included Supplementary Figure 4 and 9 to show the exact editing events in all the different editing scenarios.

7. For Fig 6, why only homozygous clones found even without MMC? Was the overall editing higher in this experiment as compared to other experiments with same editors? Or is the selective pressure for correction of both alleles higher in methylcellulose as compared to liquid culture conditions? It would be helpful to report the editing in liquid cultured cells with this donor.

We thank the reviewer for this comment.

We also observed similar % of homozygous CFCs when ABE8e used in HD CB34+ cells (Supplementary Fig. 6D and E). For this reason we think that this level of homozygosis is associated to the high editing percentage obtained using ABE8e. Gene editing in liquid culture has been now included in Fig.6C.

8. Was there an interferon response of HSPCs following mRNA/sgRNA electroporation?

We thank the reviewer for this comment.

Although interferon response has not been measured in our study, previous studies have associated the use of phosphorylated IVT sgRNA. This effect is ablated by the use of chemically modified sgRNAs (Wienert et al., 2018 & Kim et al., 2018).

9. The authors state that unbiased identification of genomic and transcriptomic off-targets are an important next step. I suggest the authors provide more details about what kind of study design they anticipate, and power to perform unbiased genomic off-target analysis, including for gRNA-independent deaminase effects.

We thank the reviewer for this comment.

We have inserted detailed experimental suggestions in the discussion section, suggesting whole genome sequencing and RNA sequencing before and after base edit experiments.

10. Why is editing at 5 days ~30% in Fig 3b but ~50% in Fig 1C (for FA-55) with seemingly same conditions?

We thank the reviewer for this comment.

Though we try to keep the same conditions for our electroporation experiments, variability among different batches of mRNA and the proliferation status of the cells can lead to this variability.

Minor:

- There are some issues with the grammar and syntax throughout, e.g. line 59, so suggest to carefully copy edit the entire manuscript.

We thank the reviewer for spotting these mistakes and we corrected grammar and syntax in the text accordingly.

- If space is limited, it would seem that some of the figures could be combined or some could be moved to supplement.

Reviewer #3 (Remarks to the Author):

Fanconi anemia (FA) is a prototypical hereditary disorder that is deficient in DNA repair, stem cell maintenance, and genome stability. The patient-derived hematopoietic stem cells might be corrected by lentiviral transduction, however, a more desirable way for gene therapy is the correction of the specific causative FA gene mutations. In this manuscript, the authors utilized two base editing systems and showed that they could correct the two FANCA mutations even in FA patient-derived hematopoietic stem/progenitor cells with excellent efficiency. As expected, the gene correction restores functional defects in the FA pathway such as MMC sensitivity.

In essence, this paper described the application of preexisting base editing technology to an important genetic disorder. The results were properly described with high standards. I have only a few comments.

We thank the referee for the enthusiasm for our manuscript and suggestions.

1. P.2, line 40. The number of the so far identified FA gene is 22, not 23 as described in the cited review (Ref 8)

We thank the reviewer for spotting the mistake and we corrected it in the text accordingly.

2. The corrected FANCA mutations are not necessarily “the most prevalent” (p.3, line 75). That depends on the ethnic group, and they should specify in which population they are most prevalent.

We thank the reviewer for spotting the inconsistency and we specified it in the text.